# Unlearning Isn't Forgetting: Revealing Hidden Leakage in Class Unlearning Evaluations

**Ali Ebrahimpour-Boroojeny** [* 1]   **Yian Wang** [* 1]   **Hari Sundaram** [1]

## Abstract

In this paper, we reveal a significant shortcoming in class unlearning evaluations: overlooking the underlying class geometry can cause information leakage about the forgotten class. We further propose a simple unlearning strategy to mitigate this issue. We introduce Class Membership Inference Attack (CMIA) that uses the probabilities assigned by the model to neighboring classes to detect unlearned samples. We find that existing unlearning methods are vulnerable to CMIA across multiple datasets. We then propose a new fine-tuning objective that mitigates this privacy leakage by approximating, for forget-class inputs, the distribution over the remaining classes that a retrained-from-scratch model would produce. To construct this approximation, we estimate inter-class similarity and tilt the target model's distribution accordingly. The resulting Tilted REWeighting (TREW) distribution serves as the desired distribution during fine-tuning. We also show that across multiple benchmarks, TREW matches or surpasses existing unlearning methods on prior unlearning metrics. More specifically, on CIFAR-10, it reduces the gap with retrained models by $19\%$ and $46\%$ for U-LiRA and CMIA scores, accordingly, compared to the SOTA method for each category. Our code is publicly available[2].

## 1. Introduction

Machine learning models deployed in real-world applications must support the ability to forget specific information upon request. Class unlearning is particularly challenging because retraining large models from scratch can be pro-

hibitively expensive and it is difficult to change a trained model to completely forget about a learned class or concept. However, privacy regulations such as the EU's GDPR (Voigt & Von dem Bussche, 2017) mandate the "right to be forgotten," enforcing model owners to delete personal data upon request. Early methods for machine unlearning primarily expedite the retraining of models at the potential cost to model performance (Bourtoule et al., 2021). These models also have high computational overhead.

Beyond retraining from scratch, prior work on machine unlearning has explored both exact and approximate approaches. Exact methods offer formal guarantees but typically rely on restrictive assumptions such as convex objectives or constrained training procedures, limiting their applicability to modern deep networks (Izzo et al., 2021; Thudi et al., 2022). As a result, most practical work focuses on approximate unlearning, including fine-tuning and weight-level interventions, which are computationally efficient but generally lack strong guarantees (Golatkar et al., 2020; Jia et al., 2023; Fan et al., 2023; Ebrahimpour-Boroojeny et al., 2025). More recently, class unlearning has emerged as a distinct setting that aims to remove an entire semantic class rather than individual samples (Warnecke et al., 2021; Poppi et al., 2024; Chen et al., 2023; Chang et al., 2024), posing new challenges for both evaluation and robustness.

The setting of class unlearning differs fundamentally from unlearning random training samples. An unlearned model must be free of any residual signal of the forgotten class, whereas merely enforcing low or zero accuracy on that class reflects only superficial forgetting rather than true unlearning. Ignoring this distinction, we show that existing methods are vulnerable to new attacks specifically designed for class-wise membership inference, such as CMIA introduced in this work. This issue has remained largely unnoticed because prior evaluations are designed for random data removal and do not account for the behavior of models retrained from scratch on all remaining classes.

We then propose a new training objective that enhances the robustness to CMIA, without computational overhead over finetuning-based unlearning methods. When a class is marked for removal, this method first reassigns its predicted probability mass proportionally to the remaining classes,

---

[1]Department of Computer Science, University of Illinois at Urbana-Champaign. Correspondence to: Ali Ebrahimpour-Boroojeny <ae20@illinois.edu>.

*Proceedings of the $43^{rd}$ International Conference on Machine Learning*, Seoul, South Korea. PMLR 306, 2026. Copyright 2026 by the author(s).

[2]https://github.com/CrowdDynamicsLab/ICML_TREW

and then tilts this distribution according to inter-class similarities to better approximate the distribution of the models retrained from scratch on the remaining classes. Our proposed method not only exhibits strong robustness against CMIA, it outperforms existing methods across prior evaluation metrics.

To evaluate our proposed approaches, we conduct comprehensive experiments on MNIST, CIFAR-10, CIFAR-100, and Tiny-ImageNet, using 10 state-of-the-art unlearning methods. In addition to common evaluations in prior work MIA (Hayes et al., 2024), we show robustness against stronger MIA (U-LiRA (Hayes et al., 2024)). Additionally, we present the results for robustness against **CMIA**, which we specifically designed for evaluation of class unlearning methods. Our results show that simple output-space interventions effectively approximates the distribution of retrained models on samples from the forget class without significant computational overhead. Our contributions can be summarized as follows:

**Class membership inference via nearest neighbors.** We propose a new membership inference attack, CMIA, that utilizes the probabilities assigned to the nearest-neighbor class of the forget class to detect whether its samples have been used for training the model. Unlike prior MIAs designed for random data removal, CMIA exposes fine-grained, class-structured leakage that persists in existing class unlearning methods. Notably, CMIA can operate without access to training data, which entails a more restrictive adversarial setting.

**Lightweight unlearning loss modification for output reweighting.** We propose Tilted REWeighting (TREW), a simple yet effective modification to fine-tuning that removes the influence of a forgotten class by approximating the output distribution of scratch-retrained models. Unlike prior methods, TREW better captures the fine-grained behavior of retrained models and is therefore more robust to CMIA and other existing attacks. This is achieved via a lightweight post-hoc probability reassignment that adjusts decision boundaries among retained classes while remaining computationally efficient.

## 2. Related Work

In this section, we review prior work on machine and class unlearning. In addition, we discuss prior work on membership inference attacks and their use cases in unlearning evaluation.

**Machine Unlearning.** Early work on machine unlearning focused on retraining models from scratch on the retained data, which is accurate but impractical for deep networks (Bourtoule et al., 2021). Another line of work focusing on certified unlearning face the same issue. Guo et al.

(2019) propose efficient certified unlearning method for $l_2$ regularized linear models (e.g., logistic regression). Izzo et al. (2021) propose certified unlearning for linear and logistic models. Sekhari et al. (2021) assume strong convexity. More recent methods, try to expand the scope of certified unlearning literature by making other assumptions, such as Lipschitz continuity of Hessian of the loss (Zhang et al., 2024). This motivated approximate unlearning methods that avoid full retraining. One line of work leverages influence functions to estimate parameter changes caused by removing specific samples, enabling efficient unlearning of features or labels under theoretical guarantees (Warnecke et al., 2021). Other approaches approximate data deletion via low-cost updates or training-trajectory analysis, including projection-based updates and SGD unrolling (Izzo et al., 2021; Thudi et al., 2022), or by pruning models prior to unlearning to reduce residual dependence on the forget set (Jia et al., 2023). Another class of methods removes targeted knowledge through fine-tuning. These include injecting noise guided by Fisher information (Golatkar et al., 2020), saliency-based weight adjustment (Fan et al., 2023), selective synaptic damping (Foster et al., 2024), instance-wise unlearning with adversarial, regularization (Cha et al., 2024), augmenting the data with adversarial examples (Ebrahimpour-Boroojeny et al., 2025), and approaches that redirect forget features toward alternative classes while preserving utility (Bonato et al., 2024). Recent work on class unlearning focuses on removing the influence of an entire class while preserving performance on the remaining data. One area of work modifies decision boundaries by fine-tuning on pseudo-class data (Chen et al., 2023). Other approaches identify and perturb class-specific representations using attribution-based methods (Chang et al., 2024), construct counterfactual samples that erase class information (Shen et al., 2024), or rely on teacher–student training with intentionally degraded teachers (Chundawat et al., 2023). Additional strategies include fast impair–repair fine-tuning (Tarun et al., 2023) and representation subtraction via low-rank decomposition to isolate class-specific features (Kodge et al., 2024).

**Membership Inference Attacks.** A membership inference attack (MIA) tests whether a specific example was part of a model's training set by exploiting the fact that overfitted models tend to behave differently on seen (members) vs. unseen points (non-members). The original MIA has a shadow-model attack that queries a target to collect confidence vectors and trains an attack classifier to decide membership (Shokri et al., 2017). Reframing evaluation toward the low-FPR regime, Carlini et al. (2022) builds a per-example likelihood ratio that improves TPR at small FPRs. And Kodge et al. (2024) proposes a low-cost, high-power MIA through a Support Vector Machine. More recently, Zarifzadeh et al. (2024) designs a robust, low-cost statistical test by composing pairwise likelihood

ratios against population draws, outperforming prior methods even with very few reference models. Also by adapting LiRA, Hayes et al. (2024) introduces per-example unlearning MIAs, showing that stronger, tailored attacks reveal overestimated privacy in prior evaluations and can even degrade retain-set privacy, urging more rigorous U-MIA testing.

# 3. Methods

To build a precise understanding of unlearning objectives and challenges, we first formalize the problem in Section 3.1. In Section 3.2, we analyze how a retrained model behaves on forgotten samples—an aspect largely overlooked in prior work—and use this insight to design a class-wise MIA in Section 3.3 that reveals vulnerabilities in existing class unlearning methods. In Section 3.4 we introduce an unlearning strategy that mitigates this shortcoming and enhances the effectiveness of class unlearning.

## 3.1. Problem Definition: Class Unlearning

Let $\mathcal{D} = \{(\mathbf{x}_i, y_i)\}_{i=1}^N$ be the full training set with the set of labels $\mathcal{Y}$ containing $K$ different classes. And let $y_f \in \mathcal{Y}$ denote the class to forget. The set of samples to forget is defined as $\mathcal{D}_f = (\mathbf{x}_i, y_i) \subset \mathcal{D} \mid_{y_i = y_f}$, and the retained set is $\mathcal{D}_r = \mathcal{D} \setminus \mathcal{D}_f$. The goal of class unlearning is to produce a model that behaves as if trained only on $\mathcal{D}_r$, i.e., with no influence from $\mathcal{D}_f$.

## 3.2. Motivation

Models retrained on $\mathcal{D}_r$ often assign skewed predictions to $\mathcal{D}_f$ based on semantic similarity. For example, see Figure 1 that shows the prediction on a ResNet18 model trained on CIFAR-10. The *Retrain* model (a model trained on $D_r$) that has never seen `automobile` samples tends to misclassify them as `truck`, which is visually and semantically similar (see Figures 3 and 10 for further empirical evidence, and Figure 6 for other examples). This behavior is not specific to model architecture but emerges from underlying data-level class similarities. This behavior has been overlooked by prior evaluation methods in class unlearning literature which focus only on the logit for the forget class.

> **Key Insight:** *The retrained models exhibit structured misclassifications for forgotten classes—typically to semantically close retained classes. Approximate methods that do not replicate this behavior are susceptible to privacy attacks.*

## 3.3. Membership Inference for Class Unlearning

Motivated by the insight in Section 3.2, we first formalize the threat model for the problem of class-wise (as opposed to sample-wise) membership inference and then propose *Class Membership Inference Attack* **CMIA**, that utilizes the probabilities assigned to the class closest to the forget class.

**Threat model.** Our threat model membership inference attack in class unlearning is inspired by the setting of U-MIA (Hayes et al., 2024) for evaluation of unlearning a random set of points. We propose the following game that instantiates the adversary:

1. Assume a model architecture $\mathcal{M}$ as well as $\mathcal{D}, \mathcal{Y}, y_f$, and $\mathcal{D}_f$ according to the definitions in Section 3.1.

2. The challenger trains $\mathcal{M}$ on $\mathcal{D}$ to derive parameters $\mathcal{M}_o$. It then applies an unlearning algorithm to the model to unlearn class $y_f$ and derive model $\mathcal{M}_u$. The challenger also trains a model on $\mathcal{D} \setminus \mathcal{D}_f$ to derive model $\mathcal{M}_t$.

3. The challenger flips a fair coin $b \in \{0, 1\}$, and sends to the adversary $(\mathcal{M}_t, y_f)$ if $b = 0$ and $(\mathcal{M}_u, y_f)$ if $b = 1$.

4. The adversary creates a decision rule $h : (\mathcal{M}, y) \to \{0, 1\}$ that predicts whether the training data for deriving $\mathcal{M}$ had included the class $y$ in it. The adversary wins if $h(\mathcal{M}, y) = b$.

For the purpose of unlearning evaluation, as is common in prior work (Chen et al., 2023; Fan et al., 2023), for the main experiments we assume that the adversary (evaluator) has access to $\mathcal{D} \setminus \mathcal{D}_f$ to train $n$ retrained (a.k.a. shadow) models. In our experiments, we set $n = 3$. In addition, in Section B.10, we show that our method can be used in a black-box setting, where the adversary has access only to test samples from the remaining classes (disjoint from $\mathcal{D}$), and it works even when using a single shadow model.

**Class Membership Inference Attack (CMIA).** We denote by $\mathcal{M}_o$ the *target* model, by $\mathcal{M}_u$ the *unlearned* model, and by $\mathcal{M}_t$ the *retrained* model. Suppose we have $n$ retrained models: $\mathcal{M}_t^{[n]} = \{\mathcal{M}_t^{(1)}, \mathcal{M}_t^{(2)}, \ldots, \mathcal{M}_t^{(n)}\}$. Let $\mathcal{R} = \{r_1, r_2, \ldots, r_{K-1}\}$ be the set of remaining classes, and let $y_f$ denote the forget class. For each class $r_i$, the test set is split to the following disjoint subsets: $D_{r_i\text{-test}}$ (i.e., test samples belonging to class $r_i$), $D_{f\text{-test}}$ (i.e., test samples belonging to the forget class), and $D_{r_{\hat{i}}\text{-test}}$ where $r_{\hat{i}} = r_j \in \mathcal{R}, j \neq i$ (i.e., the remaining of the test samples).

For a model $\mathcal{M}$ and class $r_i$, let $z_{\mathcal{M}}(x, r_i)$ denote the logit value corresponding to class $r_i$ when sample $x$ is given as input. For each remaining class $r_i$, and for each retrained model $\mathcal{M}_t^{(j)}$, construct the training data:

$$\mathcal{T}_i^{(j)} = \left\{ \left( z_{\mathcal{M}_t^{(j)}}(x, r_i), 1 \right) \mid x \in D_{r_i\text{-test}} \right\}$$
$$\cup \left\{ \left( z_{\mathcal{M}_t^{(j)}}(x, r_i), 0 \right) \mid x \in D_{r_{\hat{i}}\text{-test}} \right\} \quad (1)$$

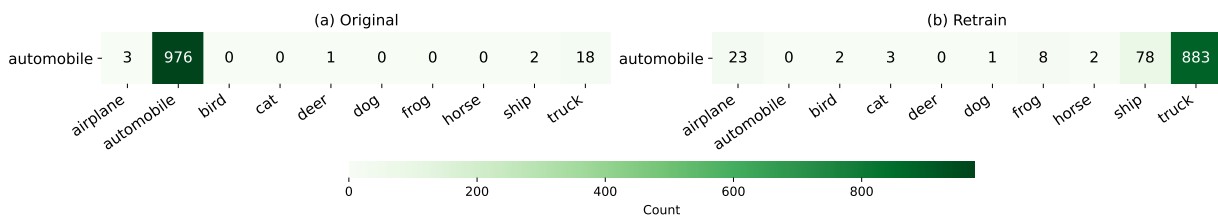

*Figure 1.* The predicted labels and their corresponding counts for samples belonging to the `automobile` class for the original model (left) and the retrained model (right). Retrained model's predictions on the forget class is skewed toward similar classes.

A binary classifier $h_i^{(j)} : \mathbb{R} \to \{0, 1\}$ is then trained to discriminate $D_{r_i\text{-test}}$ vs. $D_{r_{\hat{i}}\text{-test}}$. The accuracy of this classifier on the forget-class test data is defined as

$$\text{Acc}_i^{(j)} = \frac{1}{|D_{f\text{-test}}|} \sum_{x \in D_{f\text{-test}}} \mathbf{1}\left(h_i^{(j)}(z_{\mathcal{M}_t^{(j)}}(x, r_i)) = 1\right) \tag{2}$$

For each class $r_i$, we compute the mean accuracy across all the retrained models to derive: $\bar{\text{Acc}}_i = \frac{1}{n} \sum_{j=1}^{n} \text{Acc}_i^{(j)}$. Finally, we select the *nearest neighbor* class as:

$$r_n := \arg\max_{r_i \in \mathcal{R}} \bar{\text{Acc}}_i \tag{3}$$

To evaluate the unlearned model $\mathcal{M}_u$, we compute

$$\text{Acc}_{r_n}^{\mathcal{M}_u} = \frac{1}{|D_{f\text{-test}}|} \sum_{x \in D_{f\text{-test}}} \mathbf{1}\left(h_{r_n}^{\mathcal{M}_u}(z_{\mathcal{M}_u}(x, r_n)) = 1\right), \tag{4}$$

where $h_{r_n}^{\mathcal{M}_u}$ is trained the same way as the previous classifiers but using logits from $\mathcal{M}_u$ on $D_{r_n\text{-test}}$ and $D_{r_{\hat{n}}\text{-test}}$. We then consider the gap between $\text{Acc}_{r_n}^{\mathcal{M}_u}$ and $\bar{\text{Acc}}_{r_n}$ as a measure of unlearning effectiveness for model $\mathcal{M}_u$. In our experiments, we found that $\bar{\text{Acc}}_i$ has a low variance among different retrained models, and therefore as few as a single retrained model is enough to derive strong results. In most experiments we use three retrained models and report the standard deviation.

Table 1 shows the the value of $\bar{\text{Acc}}_i$ (the `Retrain` column) when forgetting a certain class from a ResNet18 model trained on MNIST, CIFAR-10 and CIFAR-100. As the results show, there is a large gap between $\bar{\text{Acc}}_i$ and $\text{Acc}_{r_n}^{\mathcal{M}_u}$, especially for the more recent unlearning methods that try to optimize for regular MIA score (see Table 2 and Table 4). This reveals a major shortcoming of the current evaluation methods for unlearning and the need for complementary methods such as CMIA that perform a more comprehensive analysis of the behavior of unlearned models rather than only focusing on the logit value for the forget class.

> **Main Takeaway:** *Existing SOTA methods leak membership under variants of MIAs that probe the class-wise behavior of the model, such as CMIA.*

### 3.4. Tilted REWeighting (TREW)

As mentioned in prior sections, the susceptibility of existing unlearning methods to CMIA arises from the fact that they fail to mimic the fine-grained behavior of the retrained models on samples from the forget class. A basic solution to enforce this similarity is to utilize the probabilities that the original model assigns to other classes when predicting on the forget samples. That provides us with an estimate of how the probabilities should redistribute when the forget label is enforced to be zero. More specifically, let $p(y \mid x)$ be the original model's output distribution. We perform a rescaling to remove the forget class $y_f$ and derive the reweighted distribution on the remaining classes:

$$\tilde{p}(y \mid x) = \frac{p(y \mid x)}{1 - p(y_f \mid x)} (y \neq y_f), \tilde{p}(y_f \mid x) = 0 \tag{5}$$

Using $\tilde{p}$ as the target distribution when fine-tuning on the samples of the forget class, enforces zero probability for the forget label and rescales the probability for other labels to sum up to 1. However, this assumes that the probabilities of other classes would increase proportionally in the absence of the forget class. As Figure 4 and Figure 5 in Section A.2 show, this assumption does not hold and the retrained models will have a much higher bias when predicting on the forget samples. This systematic bias arises from the fact that the forget class has different levels of similarity to other classes, and a model that has never seen the forget class, would assign higher probabilities to more similar classes.

To capture the systematic bias toward more similar classes, while adding minimal additional constraint to the target distribution, we impose a first-moment constraint given a set of similarity scores between the forget class and the remaining classes. More specifically, let $s_y$, $y \neq y_f$ show the similarity between class $y$ and $y_f$. Then the set of probability functions over the remaining classes that satisfy this constraint is:

$$\mathcal{A} := \left\{ q \in \Delta^{K-1} : \sum_{y \neq f} q(y \mid x) s_y = c \right\}, \tag{6}$$

where the scalar $c$ specifies the target expected similarity under $q(\cdot \mid x)$. This constraint reflects the empirical obser-

| Dataset | Retrain | FT | RL | GA | SalUn | BU | l1 | SVD | SCRUB | SCAR | l2ul | TREW |
|---|---|---|---|---|---|---|---|---|---|---|---|---|
| MNIST ($8 \rightarrow 3$) | 74.61 ± 1.28 | 58.67 ± 0.69 | 46.11 ± 0.88 | 19.28 ± 1.64 | 7.39 ± 2.97 | 9.23 ± 3.61 | 5.41 ± 1.85 | 49.56 ± 1.87 | 8.05 ± 2.42 | 44.28 ± 2.37 | 33.41 ± 3.66 | **71.25± 1.77** |
| CIFAR-10 (auto→ truck) | 90.10± 1.04 | 76.65 ± 1.06 | 35.18± 3.43 | 17.14± 2.19 | 6.51 ± 4.37 | 21.53 ± 1.68 | 10.95 ± 1.94 | 52.23 ± 3.89 | 9.74 ± 1.73 | 56.08± 2.04 | 24.01 ± 1.30 | **83.63± 1.48** |
| CIFAR-100 (beaver → shrew) | 74.87± 1.55 | 56.76 ± 1.04 | 11.08 ± 2.15 | 14.42 ± 1.17 | 6.53 ± 1.24 | 13.73 ± 3.25 | 4.51 ± 3.50 | 39.18 ± 1.11 | 7.54 ± 1.88 | 44.82 ± 1.46 | 19.44 ± 2.75 | **71.42 ± 1.36** |

*Table 1.* CMIA accuracy (using 3 retrained models) across three datasets. The gap with the retrained models reveals under-performance in many of the SOTA unlearning methods that have been evaluated using only regular MIAs.

vation that retrained models consistently redistribute probability mass from the forgotten class toward a small set of neighboring classes, inducing a class-level bias with low intra-class variation. Different values of $c$ correspond to different degrees of bias toward classes that are more similar to $y_f$, with larger $c$ placing greater mass on higher-similarity neighbors.

Now we need a candidate $q^*(y|x) \in \mathcal{A}$, such that it remains close to the original distribution $p(y|x)$ (and consequently to $\tilde{p}$). In Proposition 3.1 we show that our desired distribution will be a *tilted* version of $\tilde{p}(y|x)$ using the score values and has the following form:

$$q^*(y \mid x) = \frac{\tilde{p}(y \mid x)\, \exp(\beta\, s_y)}{\sum_{j \neq y_f} \tilde{p}(j \mid x)\, \exp(\beta\, s_j)}, q^*(y_f \mid x) = 0. \tag{7}$$

Here, $\beta \in \mathbb{R}$ controls the strength of the tilt and acts as the Lagrange multiplier associated with the expected-similarity constraint. Although the target value $c$ is not known a priori, varying $\beta$ traces the feasible range of expected similarities induced by $q^*$. When $\beta = 0$, $q^*$ reduces to the plain renormalized distribution $\tilde{p}$, while larger $\beta > 0$ increasingly redistributes probability mass toward classes that are more similar to $y_f$.

The intuition behind Tilted REWeighting is shown in Figure 2. In the original model (first figure from the left), the decision boundary separates class $B$ from the others. After retraining on classes $A$ and $C$, samples from $B$ are predominantly assigned to the more similar class $A$ (second figure). A basic rescaling of probabilities fails to reproduce this shift and leads to an incorrect decision boundary (third figure). In contrast, tilting the distribution using inverse Euclidean distances between class centroids as similarity scores recovers the retrained model's decision boundary.

**Theoretical results.** First, in Proposition 3.1, we show that Equation (7) is equivalent to an *information projection* of the original model's distribution onto the retained simplex under an additional linear constraint on expected similarity (i.e., $\mathcal{A}$). In other words, $q^*$ is the distribution on the remaining classes that (i) remains close to the baseline $p$ and (ii) while satisfying the bias according to class similarities $s_y$. Then in Proposition 3.2, under mild assumptions, we show that any positive and sufficiently small value of $\beta$, tilting leads to less KL divergence from the distribution of

the retrained models. See Section A.1 for all the proofs.

**Proposition 3.1.** *Let $p(\cdot \mid x) \in \Delta^K$ be the distribution of the target model for input $x$, and let $S = \{s_y\}_{y \neq f} \subset \mathbb{R}$ be fixed similarity scores with $y_f$. Given $c \in \mathbb{R}$ in the convex hull of $S$, the information projection of $p$ onto the probability simplex of retrained classes (i.e., $q(\cdot \mid x) \in \Delta^{K-1}$) with linear constraint $\sum_{y \neq f} q(y)\, s_y = c$, has form of Equation (7), where $\beta$ is some unique scalar such that $\sum_y q_\beta^\star(y \mid x)\, s_y = c$.*

Note that the unique scalar $\beta$ in 3.1 that corresponds to the first moment of the retrained model ($\beta_t$) is not known a priori. But based on our empirical results (e.g., Figures 1, 2, 3, and 4), we know that $\beta_t > 0$, meaning that the retrained model will be more biased toward more similar classes compared to the renormalized distribution $\tilde{p}$. Consider the following function:

$$F(\beta) = \text{KL}(p^t(\cdot \mid x) \,\|\, q_\beta(\cdot \mid x)), \tag{8}$$

where $p^t$ corresponds to the Retrain distribution, and therefore $F(0)$ would be the KL divergence of the renormalized distribution ($\tilde{p}$) from the Retrain distribution. Using the convexity of $F(\beta)$ (similar to the proof of Proposition 3.1), we can prove that for any positive value of $\beta$ smaller than $\beta_t$, as well as a range of sufficiently small values larger than $\beta_t$, tilting leads to smaller KL divergence from the retrained model compared to the rescaled distribution, and therefore is more effective in the unlearning task.

**Proposition 3.2.** *Let $\Delta := F(0) - F(\beta_t) > 0$. Assume there exists $v_{\max} > 0$ such that $\text{Var}_{q_\beta}(s) \leq v_{\max}$. Then for all $0 < \beta \leq \beta_t + \sqrt{2\Delta/v_{\max}}$, we have $F(\beta) \leq F(0)$.*

Our empirical ablation study on the value of $\beta$ (Table 7), verifies the theoretical observation stated in Proposition 3.2. They show less sensitivity to the choice of $\beta$ and improvement for positive values of $\beta$ as long as extreme values are avoided. In fact, in all our experiments in comparing TREW to existing methods, we fix $\beta = 10$, for various models and datasets. Still, we believe tighter bounds on the optimum value of $\beta$ could be an interesting direction for future work.

**Score function.** For the similarity scores we use cosine similarity of the weight vectors corresponding to the logits of each class (see Section A.3 for details). We have evaluated other similarity scores based on the distances in the embedding space derived from the target model (see Section B.7

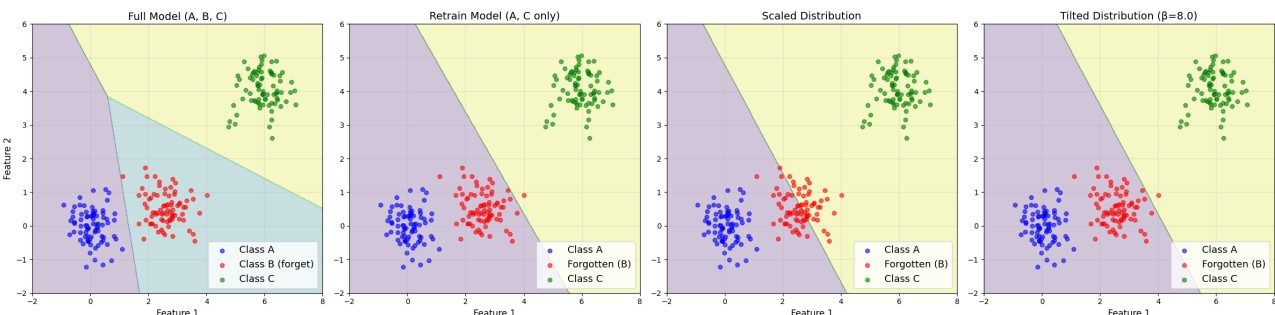

*Figure 2.* The first figure (from the left) shows the decision boundaries when class B exists. In the retrained model, $p(y_A|x)/p(y_C|x)$ would mostly increase for $x \in B$ due to the higher similarity of class A and class B (second figure). Third figure shows the decision boundary for a basic rescaling of original model's distribution, while the fourth one shows the tilted distribution, which correctly predicts the decision boundary in the retrained model.

for details). Note that we can use a sample-dependent similarity $s_y(x)$, inducing a sample-dependent first-moment constraint. While such a formulation introduces more degrees of freedom, as shown in Section B.8, we find that the sample-independent variant matches Retrain behavior more faithfully. This is because, as empirically shown in Figures 3 and 10, retrained models collapse forgotten-class representations toward a small set of neighboring classes with low intra-class variance, consistent with the low-dimensional "neural collapse" structure of trained classifiers (Papyan et al., 2020; Zhu et al., 2021). Under this structure the relevant variation is class-level rather than instance-level, and imposing sample-dependent constraints attempts to fit fine-grained variations that retraining does not produce, resulting in higher variance and reduced stability. Designing richer similarity functions that better approximate the Retrain behavior is an interesting direction for future work.

**Updated loss function.** Now that we have an approximate distribution for how the retrained model behaves on samples from the forget class, we can utilize it in our loss function for fine-tuning the model. More specifically, we fine-tune the model to minimize the following cross-entropy loss on samples from the forget class:

$$\mathcal{L}_{\text{forget}}(\mathbf{x}) = -\sum_{y=1}^{K} q^*(y \mid \mathbf{x}) \log p(y \mid \mathbf{x}). \tag{9}$$

Therefore, our new objective, Tilted REWeighting (TREW) loss, can be formulated as:

$$\min_{W} \underbrace{\sum_{(x,y)\in\mathcal{D}_{\text{retain}}} \big[ -\log p(y \mid x) \big]}_{\text{supervised loss on retained-class samples}}$$
$$+ \underbrace{\sum_{x\in\mathcal{D}_f} \big[ \mathcal{L}_{\text{forget}}(x) \big]}_{\text{reweight term on forget samples}} \tag{10}$$

In addition to TREW, we introduce TREW-2R, a lightweight variant that applies the same TREW loss but restricts gradient updates to only two randomly selected layers in the network, significantly reducing computational cost while retaining strong unlearning performance. Note that the Tilted REWeighting loss is a rather general prescription which could be utilized in many of the prior unlearning methods that perform fine-tuning on the target model. For example, some of the prior methods in unlearning focus on sparsification of the parameters that get updated during fine-tuning (Jia et al., 2023; Fan et al., 2023). Although we perform a thorough comparison to these methods, we leave further evaluations on the combination of these methods with TREW to future work.

## 4. Evaluation Setup

In Section 4.1, we elaborate on the datasets and model architectures used in our experiments. And in Section 4.2 we provide the details on the evaluations metrics. Due to space constraints, detailed descriptions of all baselines and their hyperparameter configurations are detailed in Section B.1 and Section B.2, accordingly.

### 4.1. Datasets and Models

We evaluate our method on four image datasets: **MNIST** (Deng, 2012), **CIFAR-10, CIFAR-100** (Krizhevsky et al., 2009), and **TINY-IMAGENET** (Le & Yang, 2015). For models, we use RESNET18 (He et al., 2016) and VGG19 (Simonyan & Zisserman, 2014). See Section B.2 for further details.

### 4.2. Evaluation Metrics

In Section 5.2, following the evaluation metrics used in prior work, we evaluate the unlearned models using three metrics: the accuracy on the remaining set $\text{ACC}_r$, the accuracy on the forgetting set $\text{ACC}_f$, and the Membership Inference Attack

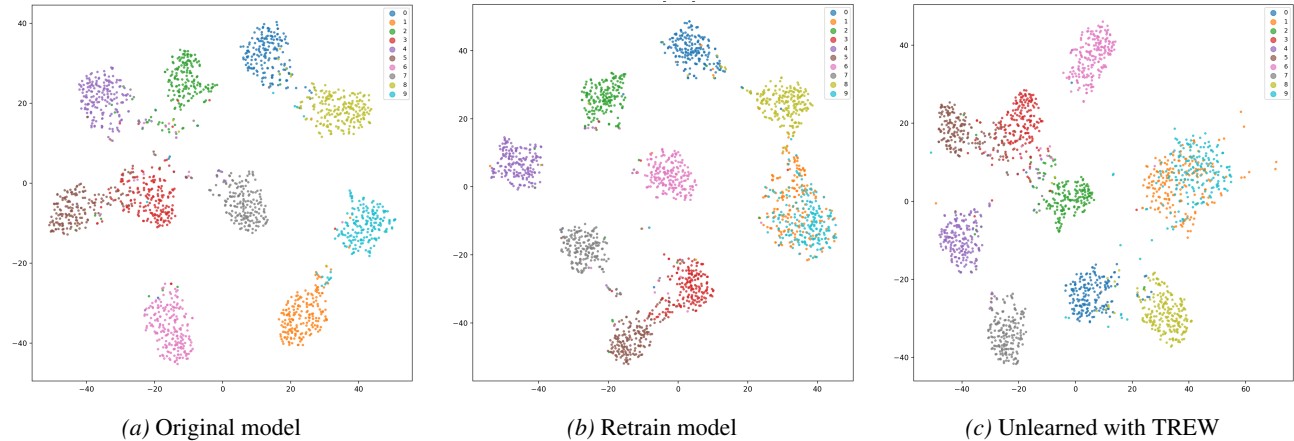

*(a)* Original model      *(b)* Retrain model      *(c)* Unlearned with TREW

*Figure 3.* The figures show the embedding of test samples (unseen during training) of CIFAR-10 derived from a ResNet18 model for the (a) original model, (b) Retrain model, and (c) the original model after applying TREW for unlearning. As the figure shows, the retrained model that has been trained without `class 1` (automobile-orange), mixes the embeddings of the samples from that class with `class 9` (truck-cyan). TREW effectively modifies the original model to replicate this behavior.

(MIA) score (Shokri et al., 2017). We applied the MIA score used by (Kodge et al., 2024). Ideally, the MIA score of the unlearned model is expected to match that of the retraining model. To perform a comprehensive comparison of the effectiveness of the unlearning methods, we utilized a SOTA MIA called U-LiRA (Hayes et al., 2024) in addition to using the MIAs from prior works. It extends the per-example Likelihood Ratio Attack (LiRA) to the unlearning setting by constructing shadow model distributions that incorporate both the training and unlearning procedures, enabling a more fine-grained, sample-specific MIA.

## 5. Results

In this section, we present a comprehensive empirical evaluation of our proposed unlearning method. In Section 5.1, we examine the effect of TREW in the representation space. In Section 5.2 we present a thorough comparison using evaluation metrics used in prior unlearning work. In Section 5.3 we present the results using a SOTA MIA method that is stronger than prior MIA methods used in unlearning evaluations and can complement the results of CMIA for a more comprehensive evaluation. We also evaluate the computation time of our method in Section B.6. The ablation study on the hyper-parameters of TREW (e.g., $\beta$) are presented in Section B.7, and the results for sample-dependent similarity scores is details in Section B.8. The results for unlearning multiple classes are in Section B.11 (for simultaneous unlearning of multiple classes) and Section B.13 (for sequential unlearning of multiple classes).

### 5.1. Effectiveness of Tilting

As a basic evaluation of the effectiveness of our method in making the original model more similar to the retrained

models, we conducted a study to evaluate the high-level geometry of the decision boundaries by looking at the class-wise clustering of samples embedding. Figure 3 shows the t-SNE plot of the CIFAR-10 embeddings derived from a trained ResNet-18 model before and after unlearning as a comparison of how these embeddings look like in retrained models. As the figures show, in the original model, the classes are well-separated in the embedding space into distinct clusters. In the retrained model, where class 1 (automobile) has been removed from the training data, the clusters corresponding to class 1 (automobile) and class 9 (Truck) have been merged. The third figure shows that our method TREW effectively replicates this behavior. Figure 10 shows a similar observation for CIFAR-100 (see Section B.9 for details).

### 5.2. Comparison to Existing Methods

Table 2 presents the results for single-class forgetting across CIFAR-10 using ResNet18 (see Table 4 in Section B.3 for VGG19). Similar results for MNIST and CIFAR-100 are shown in Section B.4 and evaluations on Tiny-ImageNet dataset for ResNet18 can be found in Section B.5. These two tables show that our method consistently achieves perfect forgetting, with $\text{ACC}_f = 0$ across all datasets and architectures. Importantly, the retained class accuracy ($\text{ACC}_r$) remains competitive with or superior to baseline unlearning methods, indicating minimal interference with the retained knowledge. Additionally, the MIA scores are either comparable to or better than prior work, suggesting that our method does not compromise membership privacy. These results demonstrate that while our approach is resilient to CMIA (see Table 1), it remains competitive to SOTA unlearning methods in common evaluation metrics used in prior work.

| Data | Method | ResNet18 (He et al., 2016) | | | | |
|------|--------|---------|---------|---------|---------|---------|
| | | $ACC_r$ ($\uparrow$) | $ACC_f$ ($\downarrow$) | $MIA$ ($\uparrow$) | $CMIA$ ($\uparrow$) | Avg gap |
| CIFAR-10 | Original | $94.74 \pm 0.09$ | $94.42 \pm 5.45$ | $0.02 \pm 0.02$ | – | – |
| | Retrain | $94.83 \pm 0.13$ | $0$ | $100 \pm 0$ | $95.27 \pm 10.92$ | $0.00$ |
| | FT (Warnecke et al., 2021) | $85.60 \pm 2.35$ | $0$ | $96.53 \pm 1.16$ | $84.78 \pm 11.57$ | $-5.80$ |
| | RL (Golatkar et al., 2020) | $84.74 \pm 4.25$ | $0$ | $94.99 \pm 1.82$ | $49.21 \pm 14.03$ | $-15.29$ |
| | GA (Thudi et al., 2022) | $90.25 \pm 0.28$ | $14.12 \pm 2.17$ | $96.70 \pm 0.10$ | $28.15 \pm 12.92$ | $-22.28$ |
| | l1 (Jia et al., 2023) | $93.21 \pm 0.13$ | $0.9 \pm 0.05$ | $100 \pm 0$ | $14.19 \pm 8.22$ | $-20.90$ |
| | BU (Chen et al., 2023) | $87.68 \pm 2.23$ | $0$ | $85.91 \pm 3.97$ | $25.73 \pm 6.17$ | $-22.70$ |
| | SalUn (Fan et al., 2023) | $92.11 \pm 0.65$ | $0$ | $96.33 \pm 2.37$ | $9.62 \pm 4.78$ | $-23.01$ |
| | SVD (Kodge et al., 2024) | $94.17 \pm 0.57$ | $0$ | $97.20 \pm 3.77$ | $67.19 \pm 17.79$ | $-7.89$ |
| | SCRUB (Kurmanji et al., 2023) | $91.07 \pm 0.79$ | $0$ | $85.01 \pm 1.02$ | $10.24 \pm 5.41$ | $-25.95$ |
| | SCAR (Bonato et al., 2024) | $93.57 \pm 0.03$ | $0$ | $95.87 \pm 3.58$ | $71.93 \pm 15.85$ | $-7.18$ |
| | l2ul (Cha et al., 2024) | $87.86 \pm 1.79$ | $0$ | $94.62 \pm 0.15$ | $32.18 \pm 8.17$ | $-18.86$ |
| | UAM (Kim et al.) | $94.15 \pm 1.91$ | $2.22 \pm 1.39$ | $100 \pm 0$ | $66.71 \pm 3.19$ | $-7.87$ |
| | **TREW** | $94.28 \pm 0.47$ | $0$ | $97.65 \pm 2.06$ | $95.82 \pm 13.72$ | $-0.59$ |
| | **TREW-2R** | $94.39 \pm 2.05$ | $0.32 \pm 0.32$ | $96.35 \pm 0.65$ | $94.67 \pm 12.47$ | $-1.09$ |

*Table 2.* Results on CIFAR-10 for ResNet18. Avg gap is the average difference across $(ACC_r, ACC_f, MIA, CMIA)$ relative to the Retrain baseline.

| Metric | TREW | TREW-2R | FT | RL | GA | SalUn | SVD | l1 | BU | SCRUB | SCAR | l2ul |
|--------|------|---------|-----|-----|-----|-------|------|-----|-----|-------|------|------|
| U-LiRA Accuracy (%) | 71.12 | **67.72** | 72.57 | 96.79 | 84.47 | 97.50 | 79.30 | 98.32 | 81.08 | 71.91 | 92.38 | 85.67 |

*Table 3.* U-LiRA Membership Inference Attack accuracy on CIFAR10 with ResNet18. Lower is better (50% indicates ideal unlearning). TREW-2R and TREW achieve the best performance.

### 5.3. Stronger MIA evaluation

We evaluate our method under a stronger MIA using the U-LiRA framework (Hayes et al., 2024), designed to assess whether unlearned models retain residual information about the forgotten class. U-LiRA simulates an adversary with access to shadow models to perform inference attacks on the forget samples.

Following the U-LiRA protocol, we train three shadow ResNet18 models on CIFAR-10. Each is unlearned with consistent hyperparameters to generate shadow unlearned models. Separately, we retrain three additional models with the forget class excluded to serve as shadow retrained models. The attacker is trained to distinguish whether a given prediction comes from an unlearned or retrained model based on class-conditional statistics, and the learned decision boundary is applied in a leave-one-out manner. A perfect unlearning method should yield 50% accuracy—indicating the accuracy of a random classifier. Based on the results reported in Table 3, our methods achieve the lowest U-LiRA accuracy, demonstrating strong resilience to adaptive attacks.

## 6. Limitations & Future Work

Further theoretical work could provide deeper insights into the effectiveness of the Tilted REWeighting objective and the design of appropriate scoring functions to better approximate the distribution of retrained models. Moreover, adding higher-order constraints could be studied for capturing the systematic bias introduced in the retrained models. While CMIA reveals shortcomings in prior work, it remains heuristic. But we believe it can serve as a great complement to other evaluation metrics for future work in class unlearning. Further improvement in this method could be another interesting avenue for exploration.

## 7. Conclusion

We introduce *Tilted REWeighting* objective for class unlearning, a lightweight technique that erases an entire class from a pretrained classifier by redistributing the forgotten class's probability mass while accounting for the systematic bias of the retrained models towards more similar classes. To evaluate how closely an unlearned model is to a fully retrained baseline, we propose *class membership-inference attack* (**CMIA**) that utilizes residual mis-mapping of forgotten outputs: previous unlearning methods that pass standard tests fail under this stronger evaluation, whereas our Tilted

REWeighting approach (TREW) remains robust. Experiments on CIFAR-10/100 and Tiny-ImageNet demonstrate that a few epochs of fine-tuning using TREW outperforms the baseline methods based on CMIA and the other existing metrics for class unlearning evaluation.

## Acknowledgments

This work used Delta computing resources at National Center for Supercomputing Applications through allocation CIS250511 from the Advanced Cyberinfrastructure Coordination Ecosystem: Services & Support (ACCESS) program (Boerner et al., 2023), which is supported by U.S. National Science Foundation grants #2138259, #2138286, #2138307, #2137603, and #2138296.

## Impact Statement

This work studies the privacy risks of class unlearning in machine learning models and proposes methods to better evaluate and mitigate residual information leakage after unlearning. By identifying systematic biases in unlearned models and introducing improved evaluation techniques, our findings may help practitioners design safer data deletion mechanisms that better align with regulatory requirements such as the right to be forgotten.

At the same time, our proposed attacks are intended solely for evaluation and auditing purposes, and could be misused if deployed irresponsibly. We therefore emphasize that these techniques should be used to strengthen privacy protections and improve unlearning methods, not to compromise user privacy. Overall, we believe this work contributes positively to the development of more reliable, transparent, and privacy-preserving machine learning systems.

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

# A. Methods (cont.)

In Section A.1 we present the proof of Proposition 3.1 and Proposition 3.2. In Section A.2 we will see further empirical observation about the necessity of using the tilted distribution to better approximate the distribution of the retrained model. In Section A.3 we will elaborate on the scoring function used in our experiments.

## A.1. Proofs

*Proof of Proposition 3.1.* We start with showing that it is enough to prove the proposition holds for $\tilde{p}$ (i.e., $q^*$ is the I-projection of $\tilde{p}$ onto the probability simplex of retrained classes with the given linear constraints).

Let $q \in \mathcal{A}$; then $q(f \mid x) = 0$ and $q$ has support only on retained labels. Note that:

$$\mathrm{KL}\big(q \,\|\, p\big) = \sum_{y \neq y_f} q(y) \log \frac{q(y)}{p(y)} = \sum_{y \neq y_f} q(y) \log \frac{q(y)}{\tilde{p}(y)(1 - p(y_f))} = \sum_{y \neq y_f} q(y) \log \frac{q(y)}{\tilde{p}(y)} \;-\; \log\big(1 - p(y_f)\big).$$

Now, the final term is independent of $q$, hence

$$\arg\min_{q \in \mathcal{A}} \mathrm{KL}\big(q \,\|\, p\big) = \arg\min_{q \in \mathcal{A}} \mathrm{KL}\big(q \,\|\, \tilde{p}\big).$$

Therefore, the information projection from $p$ onto the set $\mathcal{A}$ is equivalent to the projection from $\tilde{p}$ onto this set. Now, the objective $\mathrm{KL}(q\|\tilde{p}) = \sum_y q(y) \log\big(q(y)/\tilde{p}(y)\big)$ is strictly convex on the simplex, so any feasible minimizer is unique. Now we introduce Lagrange multipliers $\alpha, \beta$ for the normalization and expectation constraints. Stationarity with respect to $q(y)$ gives

$$\log \frac{q(y)}{\tilde{p}(y)} + 1 + \alpha + \beta s_y = 0,$$

which rearranges to

$$q(y) = \tilde{p}(y) \exp\{-1 - \alpha - \beta s_y\}.$$

Normalizing and re-parameterizing yields the exponential family

$$q_\beta(y) = \frac{\tilde{p}(y)\, e^{\beta s_y}}{\sum_j \tilde{p}(j)\, e^{\beta s_j}}.$$

It remains to pick $\beta$ so that $\sum_y q_\beta(y)\, s_y = c$. First, we define $m(\beta) = \sum_y q_\beta(y) s_y$. So we need to find value of $\beta$ that gives us $m(\beta) = c$. We suppress the fixed $x$ to lighten notation and define:

$$Z(\beta) := \sum_{j \in S} \tilde{p}(j)\, e^{\beta s_j}, \qquad q_\beta(y) := \frac{\tilde{p}(y)\, e^{\beta s_y}}{Z(\beta)},$$

Now note that:

$$Z'(\beta) = \sum_{j \in S} \tilde{p}(j)\, s_j\, e^{\beta s_j}, \qquad Z''(\beta) = \sum_{j \in S} \tilde{p}(j)\, s_j^2\, e^{\beta s_j}.$$

So, we can write:

$$m(\beta) = \sum_{y \in S} \frac{\tilde{p}(y)\, e^{\beta s_y}}{Z(\beta)}\, s_y = \frac{Z'(\beta)}{Z(\beta)}.$$

Therefore, since $Z(\beta)$ is $C^\infty$, $m(\beta)$ is $C^\infty$ as well, and by differentiating once more we get:

$$m'(\beta) = \frac{Z''(\beta)}{Z(\beta)} - \left(\frac{Z'(\beta)}{Z(\beta)}\right)^2 = \sum_{y \in S} q_\beta(y)\, s_y^2 - \left(\sum_{y \in S} q_\beta(y)\, s_y\right)^2 = \mathrm{Var}_{q_\beta}(s) \;\geq\; 0.$$

Hence $m$ is nondecreasing on $\mathbb{R}$. Moreover, if the scores are not all equal on $S$, then $q_\beta(y) > 0$ for all $y \in S$, and the variance $\mathrm{Var}_{q_\beta}(s)$ is strictly positive, so

$$\forall \beta \in \mathbb{R}: \qquad m'(\beta) = \mathrm{Var}_{q_\beta}(s) > 0,$$

i.e., $m$ is strictly increasing. Moreover, $\lim_{\beta \to -\infty} m(\beta) = \min s_y$ and $\lim_{\beta \to +\infty} m(\beta) = \max s_y$. Therefore, by the intermediate value theorem, for any feasible $c$ there exists a unique $\beta^\star$ with $m(\beta^\star) = c$. The corresponding $q^\star = q_{\beta^\star}$ is the unique minimizer. □

*Proof of Proposition 3.2.* From the proof of Proposition 3.1, recall:

$$q_\beta(y) = \frac{\tilde{p}(y)\, e^{\beta s_y}}{Z(\beta)}, \qquad Z(\beta) = \sum_{j \neq y_f} \tilde{p}(j)\, e^{\beta s_j}.$$

So we can write:

$$F(\beta) = \mathrm{KL}\big(p^t \,\|\, q_\beta\big) = \sum_{y \neq y_f} p^t(y) \log \frac{p^t(y)}{q_\beta(y)}.$$

Using $\log q_\beta(y) = \log \tilde{p}(y) + \beta s_y - \log Z(\beta)$, we can write

$$F(\beta) = \underbrace{\sum_{y \neq y_f} p^t(y) \log p^t(y)}_{\text{const}} - \underbrace{\sum_{y \neq y_f} p^t(y) \log \tilde{p}(y)}_{\text{const}} - \beta \sum_{y \neq y_f} p^t(y) s_y + \log Z(\beta) \sum_{y \neq y_f} p^t(y).$$

Using the fact that $p^t$ is a probability distribution, we know $\sum_{y \neq y_f} p^t(y) = 1$. Let $c_t := \sum_{y \neq y_f} p^t(y) s_y$ and $m(\beta)$ be as defined in the proof of Proposition 3.1. We can simplify the previous equation as:

$$F(\beta) = \text{const} - \beta c_t + \log Z(\beta).$$

Differentiating yields

$$F'(\beta) = \frac{Z'(\beta)}{Z(\beta)} - c_t = m(\beta) - c_t.$$

Differentiating once more and using the calculation from the proof of Proposition 3.1 we get:

$$F''(\beta) = m'(\beta) = \mathrm{Var}_{q_\beta}(s) \geq 0,$$

so $F$ is convex. By definition of $\beta_t$ we have $m(\beta_t) = c_t$, hence $F'(\beta_t) = 0$, i.e., $\beta_t$ is the unique minimizer of $F$.

Now fix any $\beta > \beta_t$. By Taylor's theorem with remainder, for some $\xi$ between $\beta_t$ and $\beta$,

$$F(\beta) = F(\beta_t) + F'(\beta_t)(\beta - \beta_t) + \frac{1}{2} F''(\xi)(\beta - \beta_t)^2.$$

Since $F'(\beta_t) = 0$ and $F''(\xi) = \mathrm{Var}_{q_\xi}(s) \leq v_{\max}$ by assumption, we obtain

$$F(\beta) \leq F(\beta_t) + \frac{v_{\max}}{2}(\beta - \beta_t)^2.$$

Considering $\Delta = F(0) - F(\beta_t) > 0$, if $0 < \beta - \beta_t \leq \sqrt{2\Delta/v_{\max}}$, then

$$F(\beta) \leq F(\beta_t) + \Delta = F(0).$$

Finally, since $\beta_t$ minimizes $F$, we also have $F(\beta) \leq F(0)$ for all $0 < \beta \leq \beta_t$. Combining the two cases we can conclude that for all

$$0 < \beta \leq \beta_t + \sqrt{2\Delta/v_{\max}},$$

we have $F(\beta) \leq F(0)$. □

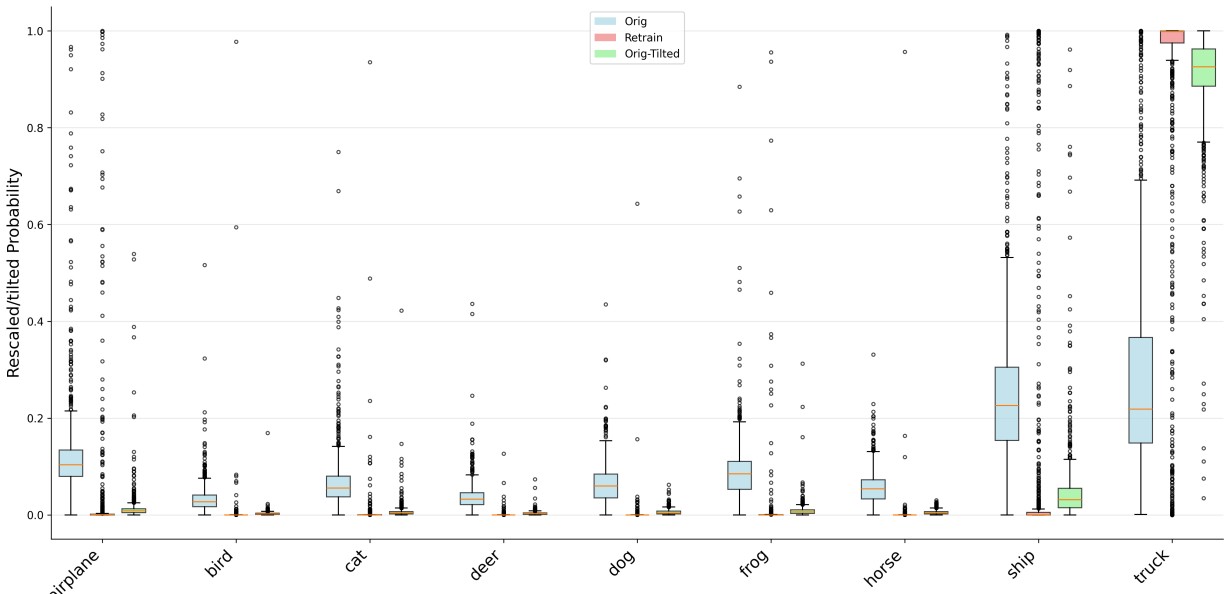

*Figure 4.* Forgetting the `automobile` class in a ResNet-18 model trained on CIFAR-10. The reweighted probabilities of the original model according to Equation (5) (`Orig`) compared to the probabilities assigned to `automobile` in the retrained model. As the figure shows, the retrained model has a much higher bias toward `Truck` class, which is better captured when using the Tilted REWeighting according to Equation (7) (`Orig-Tilted`).

## A.2. Effect of tilting

In this section, we further evaluate the insufficiency of the basic rescaling of the target model's probability distribution to approximating the distribution of the retrained model. For this purpose, we plot the conditional distributions of each remaining classes, when unlearning the `automobile` (Figure 4) and `frog` (Figure 5) classes from a ResNet18 model trained on CIFAR-10. As shown in both figures, the rescaled conditional distributions are very different from those of the retrained model. As mentioned earlier, in the retrained model, the distributions are more skewed toward a few classes. This bias toward more similar classes is better captured using the Tilted REWeighting distribution, which utilizes the inter-class similarities to adjust the rescaled distribution by introducing the bias toward more similar classes that is expected from the retrained model.

## A.3. Score function

Our goal is to assign, for each retained class $y \neq y_f$, a scalar similarity score $s_y$ that captures how likely a classifier trained *without* the forget class $y_f$ would bias predictions of $x \in D_f$ toward class $y$. We use a geometry-based score derived from the classifier's logit weights. Let $w_y \in \mathbb{R}^d$ denote the column of the final linear layer (logit weights) for class $y \in \mathcal{Y}$ in the original model. To capture the dominant inter-class structure while reducing noise and redundancy, we perform PCA on the matrix $W = [w_1 \cdots w_K] \in \mathbb{R}^{d \times K}$. Let $U_{d'} \in \mathbb{R}^{d \times d'}$ be the top $d'$ principal directions ($d' \ll d$) and define the projected class embeddings

$$\phi(w_y) \triangleq U_{d'}^\top w_y \in \mathbb{R}^{d'}.$$

We then define cosine similarities between the forget class and each retained class:

$$\tilde{s}_y \triangleq \cos\big(\phi(w_y), \phi(w_{y_f})\big) = \frac{\langle \phi(w_y), \phi(w_{y_f}) \rangle}{\|\phi(w_y)\|_2 \|\phi(w_{y_f})\|_2}, \quad y \neq y_f. \tag{11}$$

Finally, we use softmax to derive $s_y$ values in the form of probabilities from the values $\tilde{s}_y$. We use a small value for the temperature of softmax in our experiments to make the similarity values distinct for more similar classes.

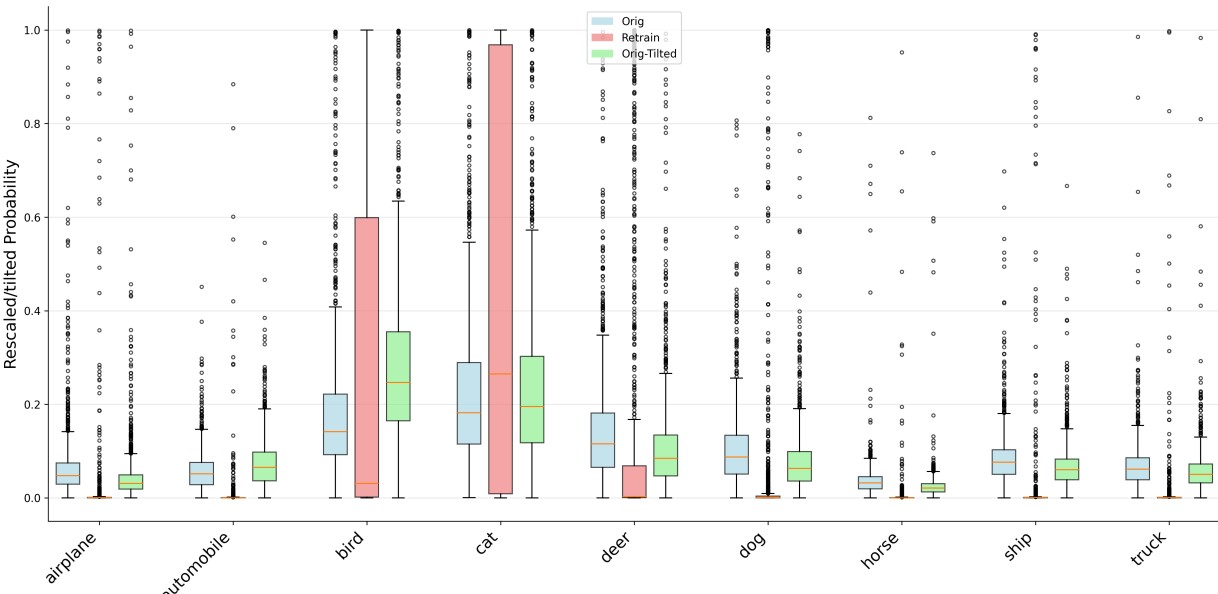

*Figure 5.* Forgetting the `frog` class in a ResNet-18 model trained on CIFAR-10. The reweighted probabilities of the original model according to Equation (5) (`Orig`) compared to the probabilities assigned to `frog` in the retrained model. As the figure shows, the retrained model has a much higher bias toward a few classes, which are better captured when using the Tilted REWeighting according to Equation (7) (`Orig-Tilted`).

# B. Experiments (cont.)

### B.1. Baseline Methods

In this section, we detail 10 state-of-the-art machine unlearning baselines.

**Retraining** refers to training a new model from scratch using only the retained dataset $\mathcal{D}_r$. This baseline serves as the ideal unlearning outcome and is commonly regarded as the "gold standard."

**Fine-tuning (FT)** (Warnecke et al., 2021) fine-tunes the source model on the remaining dataset $\mathcal{D}_r$. In contrast to this standard fine-tuning, our approach performs targeted intervention by adjusting the model's output distribution to explicitly suppress the forgotten class and reallocate its probability mass proportionally to non-forgotten classes.

**Random Labeling (RL)** (Golatkar et al., 2020) fine-tunes the model with randomly relabeled forgetting data $\mathcal{D}_f$, which helps prevent the forgotten data from influencing the model's predictions.

**Gradient Ascent (GA)** (Thudi et al., 2022) inverts SGD updates to erase the influence of specific training examples.

**l1-sparse (l1)** (Jia et al., 2023) proposes a sparsity-aware unlearning framework that prunes model weights to simplify the parameter space.

**Boundary Unlearning (BU)** (Chen et al., 2023) shifts the decision boundary of the original model to mimic the model's decisions after retraining from scratch.

**SALUN (Fan et al., 2023)** introduce the concept of weight saliency and perform unlearning by modifying the model weights rather than the entire model, improving effectiveness and efficiency.

**SVD Unlearning (SVD)** (Kodge et al., 2024) performs class unlearning by identifying and suppressing class-discriminative features through singular value decomposition (SVD) of layer-wise activations.

**SCRUB** (Kurmanji et al., 2023) proposes a novel teacher-student formulation, where the student model selectively inherit from a knowing-all teacher only when the knowledge does not pertain to the data to be deleted.

**L2UL** (Cha et al., 2024) introduces an instance-wise unlearning framework that removes information using only the pre-trained model and the data points flagged for deletion.

**SCAR** (Bonato et al., 2024) proposes a distillation-trick mechanism that transfers the original model's knowledge to the unlearned model using out-of-distribution images, preserving test performance without any retain set.

**UAM** (Kim et al.) introduces a min–max optimization framework that explicitly explores high-forget-loss regions while minimizing retain loss, enabling more effective and stable unlearning compared to fine-tuning and gradient-based baselines.

### B.2. Experiment Settings

**Datasets.** To evaluate the effectiveness of our method, we use four image datasets: **MNIST** (Deng, 2012), **CIFAR-10, CIFAR-100** (Krizhevsky et al., 2009), and **TINY-IMAGENET** (Le & Yang, 2015). For single-class forgetting, we report results averaged over different target classes with 3 random seeds for each dataset. For **MNIST** and **CIFAR-10**, the results are averaged across all 10 classes. For **CIFAR-100**, we compute the average over unlearning experiments conducted on the following 10 randomly selected classes: *apple*, *aquarium fish*, *baby*, *bear*, *beaver*, *bed*, *bee*, *beetle*, *bicycle*, and *bottle*. We use average results on multiple retrained models as an ideal reference, and assess unlearning methods by how closely they match its performance.

**Models.** MNIST and CIFAR, we use RESNET18 (He et al., 2016) and VGG19 (Simonyan & Zisserman, 2014) as the original model and for TINY-IMAGENET we use the pretrained RESNET18. Full training and hyperparameter configurations are detailed in Section B.2. All models are trained from scratch for **101 - 201 epochs** with a batch size of **128**. Optimization is performed using **stochastic gradient descent (SGD)** with a learning rate of **0.1**, **momentum of 0.9**, and **weight decay of** $5 \times 10^{-4}$. The learning rate follows a `torch.optim.lr_scheduler.StepLR` schedule with `step_size = 40` and `gamma = 0.1`.

For our unlearning procedure, we run updates for **10 epochs** with a learning rate of **0.001**. Baseline re-training budgets mirror prior work: GA (Thudi et al., 2022), FT (Warnecke et al., 2021), l1 (Jia et al., 2023), and L2UL (Cha et al., 2024) baselines are run for **20 epochs**, SVD (Kodge et al., 2024), and SCRUB (Kurmanji et al., 2023) are run for **10 epochs**, while SalUn (Fan et al., 2023) is run for **15 epochs** and SCAR (Bonato et al., 2024) for **25 epochs**. All experiments are implemented in **Python 3.11** and executed on four **NVIDIA A40** GPUs, and repeated with three different random seeds; we report the average results across those runs.

### B.3. More results on CIFAR-10

In Figure 6 we see the behavior of a models that are retrained from scratch by excluding any of the classes. This further shows how the retrained models are assign the forget samples to a few of the similar classes.

In section 5.2 we presented the results for ResNet-18 models trained on CIFAR-10. In Table 4 we present similar results for VGG19 models. As the table shows, our proposed methods outperform existing unlearning methods.

### B.4. Results on MNIST and CIFAR-100

To study performance on a higher-variance image domain, we further evaluate on **MNIST** and **CIFAR-100** with VGG and ResNet-18 backbones. As shown in Table 5, our methods deliver (near-)perfect forgetting with competitive retained accuracy and consistently stronger resistance to MIA attacks than the baselines.

### B.5. Results on Tiny-ImageNet-200

To evaluate unlearning on a more challenging benchmark, we also apply our method to the **Tiny-ImageNet-200** dataset using a ResNet-18 backbone. We average the results over 10 semantically diverse classes: *goldfish*, *European fire salamander*, *bullfrog*, *tailed frog*, *American alligator*, *boa constrictor*, *trilobite*, *scorpion*, *black widow spider*, and *tarantula*. Due to computational constraints, we compare against a selected subset of representative baselines. From Table 6, we observe that our methods achieve perfect forgetting with strong retained accuracy and MIA robustness.

### B.6. Running Time Analysis

We compare the wall-clock training time of each unlearning method on the CIFAR-100 dataset using a ResNet-18 backbone, measured on four NVIDIA A40 GPUs. As shown in Figure 7, our proposed **TREW-2R** method is among the fastest, completing in 18.25s, which is significantly faster than standard retraining-based baselines like FT and RL.

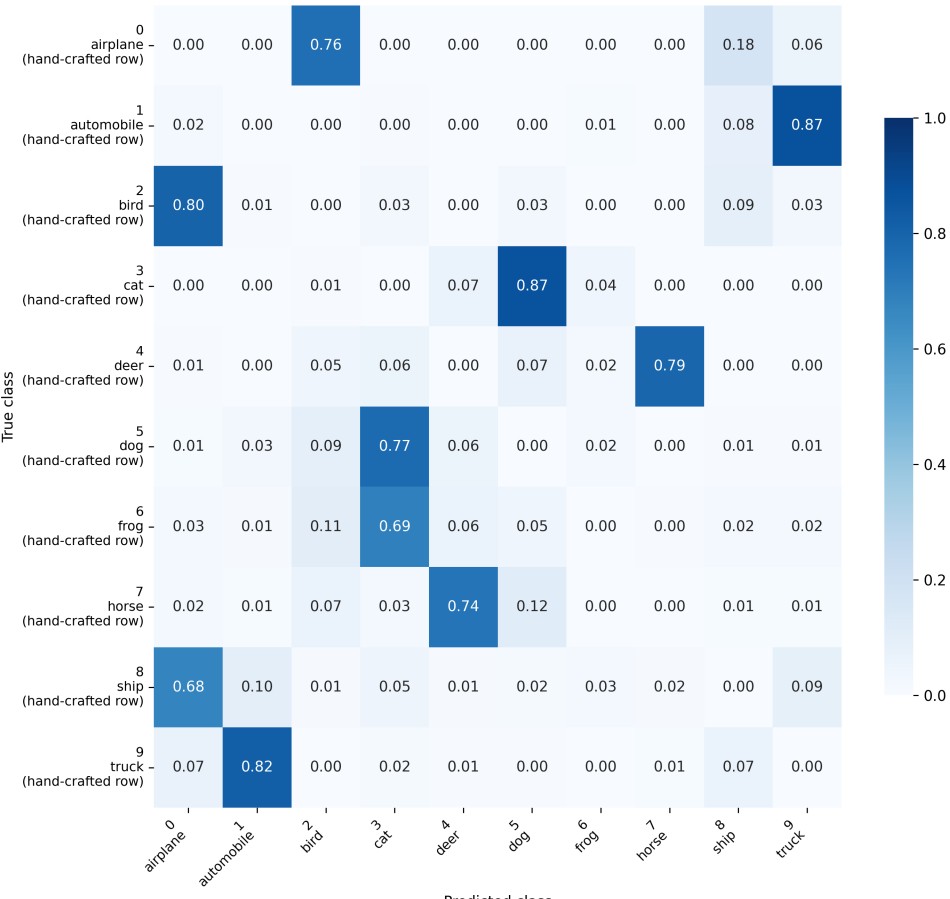

*Figure 6.* Prediction of the models (column) retrained from scratch with any of the classes (row) from CIFAR-10.

| Data | Method | VGG19 (Simonyan & Zisserman, 2014) | | | | |
|---|---|---|---|---|---|---|
| | | $ACC_r$ (↑) | $ACC_f$ (↓) | $MIA$ (↑) | $CMIA$ (↑) | Avg gap |
| CIFAR-10 | Original | $92.68 \pm 0.05$ | $92.14 \pm 6.80$ | $0$ | $-$ | $-$ |
| | Retrain | $93.45 \pm 0.10$ | $0$ | $100 \pm 0$ | $94.10 \pm 9.66$ | $0.00$ |
| | FT (Warnecke et al., 2021) | $89.33 \pm 1.86$ | $0$ | $96.94 \pm 0.91$ | $79.69 \pm 12.81$ | $-5.40$ |
| | RL (Golatkar et al., 2020) | $85.57 \pm 1.29$ | $0$ | $94.07 \pm 0.90$ | $52.11 \pm 17.14$ | $-13.95$ |
| | GA (Thudi et al., 2022) | $87.26 \pm 0.30$ | $14.4 \pm 1.59$ | $97.10 \pm 0.87$ | $27.36 \pm 10.65$ | $-15.36$ |
| | l1 (Jia et al., 2023) | $90.12 \pm 2.18$ | $0.3 \pm 0.03$ | $92.61 \pm 7.23$ | $11.44 \pm 6.39$ | $-23.27$ |
| | BU (Chen et al., 2023) | $87.32 \pm 3.08$ | $0$ | $85.94 \pm 3.71$ | $18.92 \pm 4.03$ | $-23.84$ |
| | SalUn (Fan et al., 2023) | $89.76 \pm 1.00$ | $0$ | $97.35 \pm 0.65$ | $8.59 \pm 4.01$ | $-22.96$ |
| | SVD (Kodge et al., 2024) | $91.34 \pm 0.45$ | $0$ | $98.10 \pm 1.90$ | $65.42 \pm 11.54$ | $-8.17$ |
| | SCRUB (Kurmanji et al., 2023) | $89.84 \pm 1.16$ | $0$ | $84.28 \pm 1.91$ | $14.07 \pm 6.93$ | $-24.84$ |
| | SCAR (Bonato et al., 2024) | $92.59 \pm 1.80$ | $0$ | $98.06 \pm 2.11$ | $69.51 \pm 13.45$ | $-6.85$ |
| | l2ul (Cha et al., 2024) | $89.15 \pm 0.75$ | $0$ | $96.82 \pm 1.45$ | $35.90 \pm 9.46$ | $-16.42$ |
| | **TREW** | $91.58 \pm 0.23$ | $0$ | $99.26 \pm 0.74$ | $94.63 \pm 13.34$ | $-0.52$ |
| | **TREW-2R** | $91.91 \pm 0.63$ | $0$ | $99.52 \pm 0.74$ | $93.58 \pm 14.82$ | $-0.63$ |

*Table 4.* Results on CIFAR-10 for VGG19. Avg gap is the average difference across $(ACC_r, ACC_f, MIA, CMIA)$ relative to the retrain baseline.

| Data | Method | VGG19 (Simonyan & Zisserman, 2014) | | | ResNet18 (He et al., 2016) | | |
|---|---|---|---|---|---|---|---|
| | | $ACC_r$ (↑) | $ACC_f$ (↓) | $MIA$ (↑) | $ACC_r$ (↑) | $ACC_f$ (↓) | $MIA$ (↑) |
| MNIST | Original | 99.52 ± 0.01 | 99.57 ± 1.33 | 0 | 99.65 ± 0.04 | 99.91 ± 1.05 | 0.23 ± 0.23 |
| | Retraining | 99.54 ± 0.03 | 0 | 100 ± 0 | 99.64 ± 0.05 | 0 | 100 ± 0 |
| | FT (Warnecke et al., 2021) | 99.43 ± 0.72 | 0 | 99.25 ± 0.11 | 98.16 ± 0.65 | 0 | 95.78 ± 1.28 |
| | RL (Golatkar et al., 2020) | 99.04 ± 0.19 | 0 | 99.66 ± 0.21 | 99.33 ± 0.25 | 0 | 99.64 ± 0.15 |
| | GA (Thudi et al., 2022) | 97.69 ± 2.42 | 0 | 96.86 ± 2.87 | 98.26 ± 0.12 | 14.94 ± 0.03 | 85.19 ± 0.17 |
| | l1 (Jia et al., 2023) | 94.07 ± 4.48 | 0.01 ± 0.02 | 92.55 ± 1.53 | 93.47 ± 1.98 | 0.04 ± 0.02 | 97.51 ± 0.42 |
| | BU (Chen et al., 2023) | 93.14 ± 8.19 | 0 | 95.40 ± 0.06 | 94.12 ± 6.51 | 0 | 98.62 ± 0.15 |
| | SalUn (Fan et al., 2023) | 99.23 ± 0.18 | 0 | 100 ± 0 | 99.43 ± 0.77 | 0 | 100 ± 0 |
| | SVD (Kodge et al., 2024) | 99.16 ± 0.20 | 0 | 100 ± 0 | 99.37 ± 0.32 | 0 | 99.87 ± 0.13 |
| | SCRUB (Kurmanji et al., 2023) | 99.34 ± 0.03 | 0 | 98.73 ± 0.15 | 99.45 ± 0.06 | 0 | 91.88 ± 0.34 |
| | SCAR (Bonato et al., 2024) | 98.93 ± 0.10 | 0 | 100 ± 0 | 99.20 ± 0.24 | 0 | 97.82 ± 0.68 |
| | l2ul (Cha et al., 2024) | 99.15 ± 0.12 | 0 | 100 ± 0 | 95.75 ± 0.16 | 0 | 93.83 ± 0.19 |
| | **TREW** | **99.52 ± 0.07** | **0** | **100 ± 0** | 99.49 ± 0.04 | 0 | 100 ± 0 |
| | **TREW-2R** | 99.48 ± 0.13 | 0 | 99.78 ± 0.22 | 99.45 ± 0.08 | 0.04 ± 0.06 | 99.98 ± 0.15 |
| CIFAR-100 | Original | 69.87 ± 0.80 | 70.72 ± 6.41 | 0.45 ± 0.85 | 78.52 ± 0.58 | 78.93 ± 5.77 | 0.3 ± 0.5 |
| | Retraining | 69.54 ± 0.92 | 0 | 100 ± 0 | 78.30 ± 0.84 | 0 | 100 ± 0 |
| | FT (Warnecke et al., 2021) | 65.26 ± 1.58 | 0 | 87.41 ± 3.32 | 73.37 ± 1.42 | 0 | 86.09 ± 0.46 |
| | RL (Golatkar et al., 2020) | 58.83 ± 3.04 | 0 | 87.20 ± 3.11 | 70.97 ± 1.68 | 0 | 88.65 ± 0.33 |
| | GA (Thudi et al., 2022) | 62.34 ± 0.77 | 0 | 83.58 ± 0.41 | 72.05 ± 0.19 | 0 | 85.11 ± 1.45 |
| | l1 (Jia et al., 2023) | 57.28 ± 1.47 | 0 | 83.89 ± 1.78 | 72.30 ± 1.84 | 0 | 84.75 ± 2.06 |
| | BU (Chen et al., 2023) | 59.27 ± 3.23 | 0 | 84.32 ± 2.45 | 67.52 ± 1.86 | 0 | 81.62 ± 0.79 |
| | SalUn (Fan et al., 2023) | 63.58 ± 3.06 | 0 | 99.97 ± 0.03 | 72.11 ± 1.37 | 0 | 98.65 ± 1.35 |
| | SVD (Kodge et al., 2024) | 68.53 ± 1.78 | 0 | 100 ± 0 | 75.86 ± 1.79 | 0 | 99.30 ± 0.08 |
| | SCRUB (Kurmanji et al., 2023) | 58.91 ± 1.32 | 0 | 89.01 ± 1.34 | 76.92 ± 1.06 | 0 | 87.65 ± 2.18 |
| | SCAR (Bonato et al., 2024) | 65.97 ± 1.88 | 1.38 ± 1.04 | 100 ± 0 | 75.42 ± 1.17 | 0.5 ± 0.43 | 97.61 ± 0.81 |
| | l2ul (Cha et al., 2024) | 66.77 ± 0.84 | 0 | 99.52 ± 0.53 | 71.56 ± 1.38 | 0 | 97.05 ± 1.97 |
| | **TREW** | 69.36 ± 0.61 | 0 | 99.15 ± 0.75 | **78.05 ± 0.43** | **0** | **99.56 ± 1.75** |
| | **TREW-2R** | 69.07 ± 1.69 | 0 | 99.00 ± 1.36 | 77.97 ± 0.78 | 0.01 ± 0.07 | 98.87 ± 1.15 |

*Table 5.* **Single-class forgetting on MNIST and CIFAR-100** We bold the method with the highest retained accuracy (ACC$_r$), membership attack robustness (MIA), and lowest forgotten class accuracy (ACC$_f$). Our method (TREW, TREW-2R) consistently achieves perfect forgetting (ACC$_f$ = 0), while preserving high retained accuracy and strong MIA robustness across datasets and architectures.

| Method | ResNet-18 | | |
|---|---|---|---|
| | $ACC_r$ (↑) | $ACC_f$ (↓) | $MIA$ (↑) |
| Original | 63.49 | 64.52 | 0 |
| Retrain | 63.32 | 0 | 100 |
| FT (Warnecke et al., 2021) | 58.92 | 0 | 83.62 |
| RL (Golatkar et al., 2020) | 41.58 | 0 | 83.55 |
| SCRUB (Kurmanji et al., 2023) | 62.31 | 0 | 87.73 |
| SCAR (Bonato et al., 2024) | 61.48 | 0.63 | 97.29 |
| l2ul (Cha et al., 2024) | 61.42 | 0 | 96.18 |
| TREW | 62.78 | 0 | 100 |
| **TREW-2R** | **62.84** | **0** | **100** |

*Table 6.* **Single-class forgetting on Tiny-ImageNet-200 with ResNet-18.** We report retained accuracy ($ACC_r$), forget accuracy ($ACC_f$), and membership inference attack (MIA).

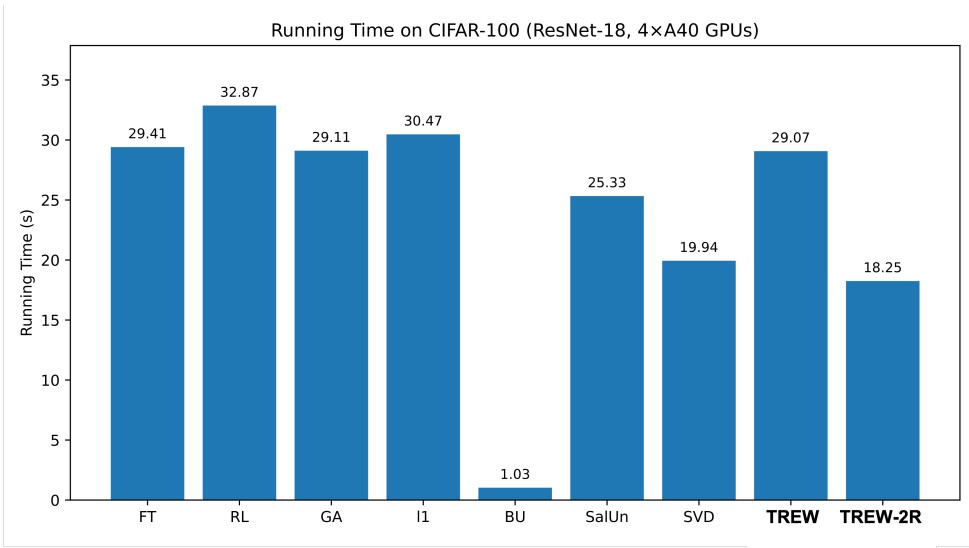

*Figure 7.* Running time comparison (in seconds per epoch) on CIFAR-100 with ResNet-18 using 4×A40 GPUs. Our TREW-2R and TREW are among the fastest methods.

Importantly, both of our methods — TREW, and TREW-2R — achieve optimal unlearning performance within just **10 epochs**. This makes them highly efficient and scalable in practice, offering strong privacy guarantees at a fraction of the computational cost of full retraining.

### B.7. Ablation Study

| $\beta_f$ | $ACC_r$ ($\uparrow$) | $ACC_f$ ($\downarrow$) | $MIA$ ($\uparrow$) | $CMIA$($\uparrow$) |
|---|---|---|---|---|
| 0 | 94.35 | 0 | 99.94 | 47.2 |
| 5 | 94.37 | 0 | 98.94 | 76.1 |
| 10 | 94.28 | 0 | 97.65 | 82.1 |
| 15 | 93.78 | 0 | 97.49 | 87.8 |
| 20 | 77.67 | 0 | 98.72 | 98.6 |

*Table 7.* Effect of $\beta_f$ on retained accuracy ($ACC_r$), forgetting accuracy ($ACC_f$), and membership inference resistance ($MIA$ and $CMIA$). Increasing $\beta_f$ improves unlearning robustness but may harm retained performance. We select $\beta_f = 10$ as it provides the best balance between maintaining accuracy on retained data and achieving strong resistance to membership inference attacks.

We explore different choices of $\beta$ as it directly affects the trade-off between retained accuracy and unlearning robustness and the results are shown in Table 7. We find that $\beta = 10$ provides the most balanced result across metrics. At this setting, the retained accuracy ($ACC_r$) remains very high (94.28), almost identical to $\beta = 0$ and $\beta = 5$, indicating that the model preserves strong predictive performance on the retained data. Meanwhile, the unlearning effectiveness improves significantly: the $MIA$ score decreases to 97.65, and the $CMIA$ score rises to 82.1. This demonstrates that the model is more resistant to membership inference attacks compared to smaller $\beta$ values, where adversaries can more easily detect forgotten samples. Although $\beta = 20$ achieves even stronger robustness ($CMIA = 98.6$), it severely degrades retained accuracy (dropping to 77.67). Therefore, we select $\beta = 10$ as it achieves the optimal balance between maintaining accuracy on retained data and providing strong attack robustness.

In addition to the ablation study on the tilt parameter ($\beta$), we performed an ablation study on the other hyper-parameters of our method, including the choice of the class-wise similarity function, inverse temperature for deriving the class-wise similarity scores, and the number of dimensions in PCA. The results are presented in Table 8.

Our experiments showed that the use of cosine similarity of class-wise weight vectors in the last layer (82.1 in CMIA) leads to empirically stronger and closer to the retrain ideal (90.1) compared to the Euclidean distances (77.8). It also shows that our model is robust to slight changes in PCA dimension, tilt parameter ($\beta$), and inverse temperature in computing the

| Hyperparameter | Value | ACC$_r$ ↑ | ACC$_f$ ↓ | MIA↑ | CMIA↑ |
|---|---|---|---|---|---|
| retrain | - | 94.58 | 0 | 100 | 90.1 |
| PCA dim | - | 93.75 | 0 | 99.12 | 86.8 |
| | 16 | 94.22 | 0 | 99.82 | 83.5 |
| | 32 | 94.27 | 0 | 97.65 | 82.1 |
| | 64 | 94.22 | 0 | 99.82 | 83.5 |
| Distance | Euclidean | 94.27 | 0 | 100 | 77.8 |
| | Cosine | 94.28 | 0 | 97.65 | 82.1 |
| Inv$_{temp}$ | 0.5 | 94.30 | 0 | 99.85 | 81.4 |
| | 1 | 94.32 | 0 | 99.12 | 81.0 |
| | 5 | 94.27 | 0 | 98.07 | 82.1 |
| | 10 | 91.85 | 0 | 97.88 | 86.1 |
| | 100 | 75.72 | 0 | 97.87 | 95.3 |

*Table 8.* Ablation study in forgetting the class `automobile` in CIFAR10. Unless otherwise noted, TREW uses cosine similarity of class-wise weight vectors as the class-wise similarity function and uses softmax with inverse temperature of 5 to derive the similarity scores. Default dimension for PCA is 32, and we use $\beta$=10.

similarity scores. Although we fixed our hyper-parameters in all our experiments (different models, datasets, and forget classes) to: a) cosine similarity of the class-wise weight vectors, b) PCA dimension of 32, c) inverse temperature of 5, and d) $\beta = 10$, as the results show, hyper-parameter tuning can lead to slight improvements in some cases.

In addition to this regular ablation study, we conducted another experiment to show the effect of inverse temperature (for computing the similarity scores) and the tilt parameter ($\beta$) in our method when applied to unlearning the `automobile` class from a ResNet-18 model trained on CIFAR-10. In this experiment, we are interested in finding out how the weight given to the nearest-neighbor class (`truck`) in equation 2 changes. Basically, we compute $\frac{\exp(\beta\,s_y)}{\sum_{j \neq y_f} \exp(\beta\,s_j)}$ for the `truck` class. Note that for the rescaled distribution given in equation 1 (which is equivalent to setting $\beta = 0$ in equation 2), this value is equal to $\frac{1}{9} \approx 0.11$. As Figure 8 shows, for the default parameters used in our experiments ($\beta = 10$ and an inverse temperature of 5), this leads to a modest boost in the weight of the most similar class (0.15). This modest boost is due to the fact that the cosine similarity values that we compute for the class-wise weight vectors lead to very small differences for some of the classes. For example, for the `automobile` class, the cosine similarities with the three most similar classes `truck`, `ship`, and `airplane` are 0.937, 0.922, and 0.919, accordingly. Therefore, as shown in the figure and the presented table, our method is robust to modest changes in the value of $\beta$ and the inverse temperature and only deteriorates at extreme values.

Also note that a benefit of using the initial softmax for computing the similarities is that despite the choice of the similarity metric and the range of values, it converts the similarity values to probabilities in range $[0, 1]$ and therefore prevents extreme changes from appearing when computing the weights in the tilted distribution of equation 2.

### B.8. Similarity Scores: Sample-dependent vs. Sample-independent

The similarity definitions in Section A.3 are *sample-independent*: for a fixed forget class $y_f$, each retained class $y \neq y_f$ receives a single score that is shared across all forget examples $x \in D_f$. Here, we implement a *sample-dependent* alternative, **TREW-SD**, where the similarity vector is computed separately for each forget example $x \in D_f$ based on distances in the model's representation space.

**Sample-dependent similarity from prototype distances (TREW-SD).**   Let $h(x) \in \mathbb{R}^d$ denote the penultimate-layer representation of the original model. We apply PCA (using the same PCA object as in Section A.3) to obtain $\psi(h(x))$. We compute per-class prototypes (means) from the training set:

$$\mu_y \triangleq \frac{1}{|D_y|} \sum_{(x_i, y_i) \in D_y} \psi(h(x_i)), \tag{12}$$

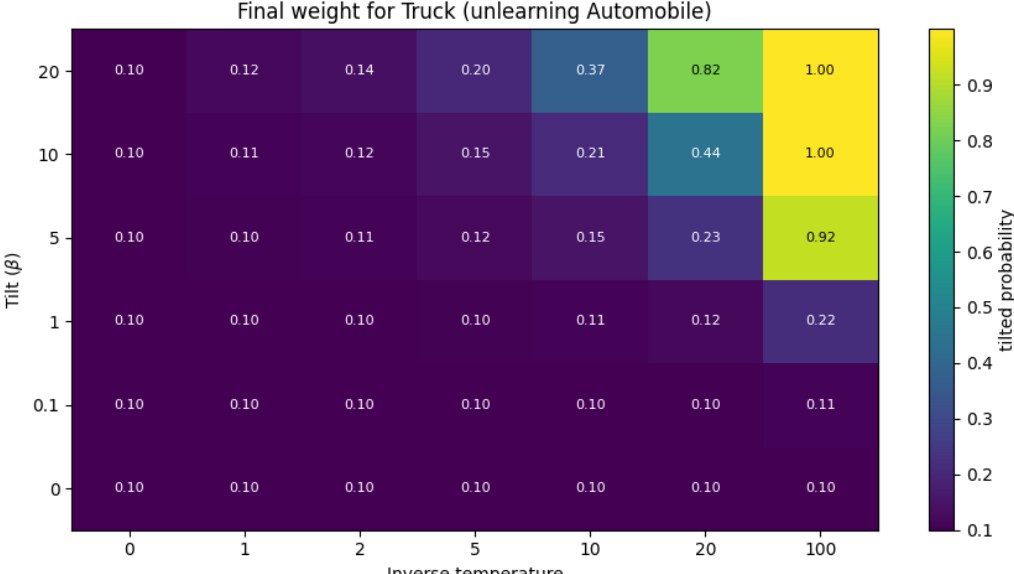

*Figure 8.* To see the actual effect of tilting the distribution based on the value of the inverse temperature and the tilt parameter ($\beta$), we plot the final weight applied to the conditional distribution $\tilde{p}(y \mid x)$ in equation 2 when $y$ is the `Truck` label. As the figure shows, for the default values of $\beta = 10$ and inverse temperature of 5 (fixed in our experiments), the weights are not very sharp and deviate slightly from the default $0.11$ ($\frac{1}{9}$) weights assigned to each conditional probability in equation 1 (the rescaled probabilities). As shown in the figure and also the table above on the ablation studies, slight changes to the tilt parameter or inverse temperature do not make large differences.

| Method | $\mathbf{ACC_r}\uparrow$ | $\mathbf{ACC_f}\downarrow$ | $\mathbf{MIA}\uparrow$ | $\mathbf{CMIA}\uparrow$ |
|---|---|---|---|---|
| Retrain (gold) | 94.58 | 0 | 100 | 90.1 |
| TREW (sample-indep.) | 94.27 | 0 | 97.65 | 82.1 |
| **TREW-SD** (sample-dep.) | 93.43 | 0 | 93.91 | 65.6 |

*Table 9.* **Sample-dependent similarity (TREW-SD).** We compare TREW using sample-independent similarity vs. TREW-SD using per-sample prototype-distance similarities (Equation (13)) under the same hyperparameters and evaluation protocol. Lower is better for privacy attack success (e.g., U-LiRA / CMIA), and higher is better for retained accuracy.

where $D_y$ denotes training samples of class $y$. For each forget sample $x \in D_f$, we compute distances to all class prototypes,

$$d_y(x) \triangleq \|\psi(h(x)) - \mu_y\|_2, \qquad y \neq y_f, \tag{13}$$

and convert these distances into a per-sample similarity distribution with separate temperature controls for the forget class vs. remaining classes.

Table 9 reports the results of this sample-dependent similarity (denoted TREW-SD) versus the cosine-based similarity (TREW) under the same evaluation protocol. As the results show, using a sample-independent similarity leads to better results than the fine-grain sample-dependent similarity measure.

## B.9. More Analysis on CIFAR-100

Figure 9 summarizes the top classes that *willow tree* samples were reassigned to after unlearning. Notably, a significant proportion of the reassigned predictions fall into semantically or visually similar categories, such as *palm tree* (25%), *forest* (17%), and *oak tree* (14%). This suggests that although the model effectively forgets the target class, it redistributes the predictions to classes with similar visual features.

Following the same procedure as in Figure 3 in Section 5.1, We perform similar analysis for CIFAR-100. We visualize the embeddings of test samples (unseen during training) derived from a ResNet18 model using t-SNE. We compare the original model, the retrained model with the target class removed, and the model unlearned using TREW.

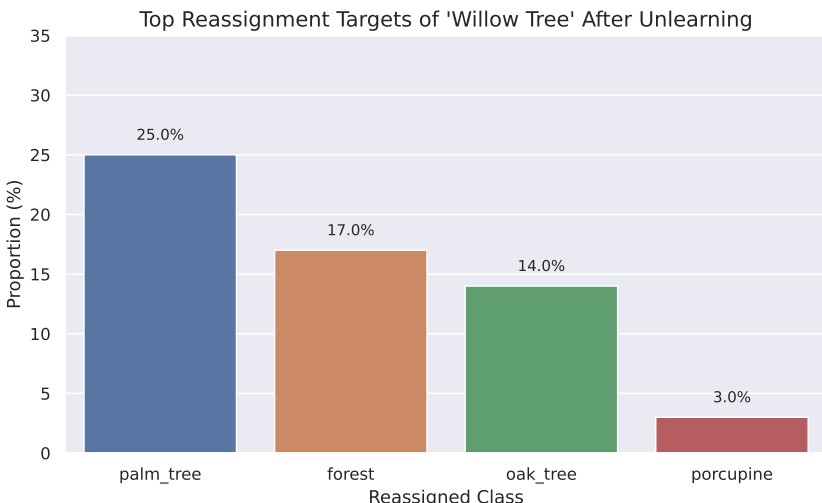

*Figure 9.* **Top reassignment targets of the forgotten class *willow tree*.** The model tends to reassign samples to semantically similar classes such as *palm tree*, *forest*, and *oak tree*.

In Figure 10, we consider a forgetting setting where the target class is class 4 (*beaver*). To make the visualization interpretable, we restrict the plot to a subset of classes while preserving the relevant semantic structure. Specifically, we include the forgotten class, its nearest neighbor class identified over all CIFAR-100 classes (class 74, *shrew*), and first 8 classes in CIFAR-100.

Consistent with our observations in Figure 3, retraining causes samples from the forgotten class to merge with semantically similar classes in the embedding space. In contrast to naive probability rescaling, TREW effectively modifies the original model to replicate this behavior, producing embeddings that closely match those of the retrained model.

### B.10. Blackbox Setting

Many prior MIAs adapted to the setting of unlearning for evaluations assume that the adversary has also access to the training data $(\mathcal{D} - \mathcal{D}_f)$ to derive logits values of the model on the training samples or train shadow models for the retrained model (Fan et al., 2023; Chen et al., 2023). However, our proposed attack also works in the setting of black-box attack where we drop this assumption and utilize the shadow retrained models trained by the adversary on public data (disjoint from the training set).

To perform this experiment and showcase the performance of CMIA in a black-box setting, we randomly split the CIFAR-10 training set into two parts. We use only the first part for training the original model. The adversary has access to the second part, which is disjoint from the training samples and can be considered public data in this setting, and uses that for training one retrained model. Then the adversary utilizes the CIFAR-10 test set (which is again considered public data) and the output probabilities from the original model to evaluate whether the original model has been trained on the forgotten class. Therefore, in this setting, we have shown the use case of CMIA in a black-box setting. Table 10 shows the results of black-box CMIA for revealing the current shortcomings in existing unlearning methods. As the results show, CMIA in the black-box setting is also very effective.

| Dataset | Retrain | FT | RL | GA | SalUn | BU | l1 | SVD | SCRUB | SCAR | l2ul | TREW |
|---|---|---|---|---|---|---|---|---|---|---|---|---|
| CIFAR-10 (auto→ truck) | 83.1 | 66.8 | 26.1 | 12.8 | 2.11 | 11.6 | 9.2 | 47.9 | 9.4 | | 55.2 | 21.5 | **86.1** |

*Table 10.* CMIA accuracy in the black-box setting. Lower gap with the retrained models indicates more effective unlearning. The gap with the retrained models reveals under-performance in many of the SOTA unlearning methods that have been evaluated using only regular MIAs.

If a public checkpoint for a model that has been trained on only the remaining classes is available to the adversary, the adversary would be able to utilize that model as it provides a similar setting to the one we used for our black-box setting (no

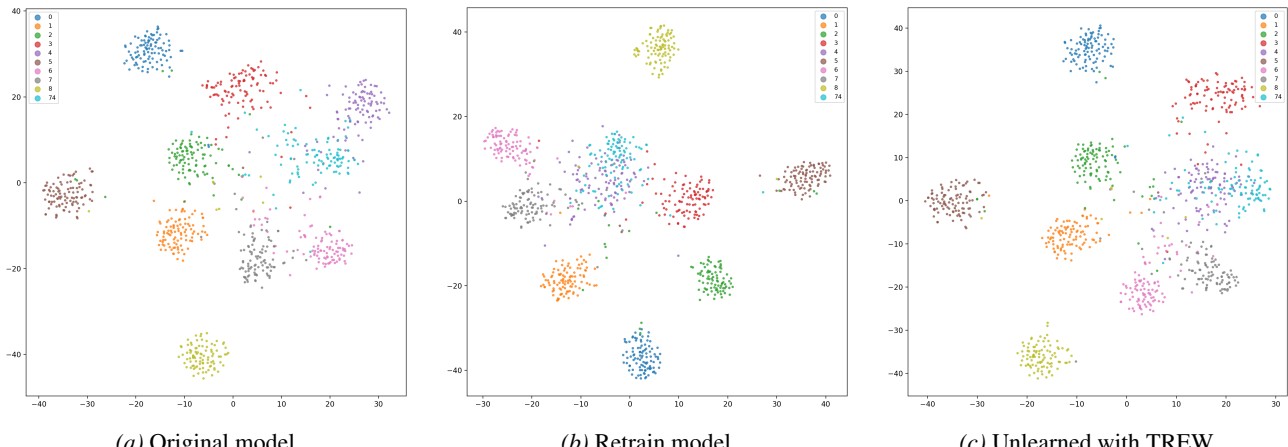

*(a)* Original model        *(b)* Retrain model        *(c)* Unlearned with TREW

*Figure 10.* The figure shows the embedding of test samples (unseen during training) of CIFAR-100 derived from a ResNet18 model for the (a) original model, (b) Retrain model, and (c) the original model after applying TREW for unlearning. As the figure shows the retrained model that has been trained without `class 4` (beaver), mixes the embeddings of the samples from that class with `class 74` (shrew). TREW effectively modifies the original model to replicate this behavior.

access to the training data and the original model parameters). Still, this new black-box experiment, which requires only one retrained model on public data to reveal the shortcomings of existing unlearning methods is a rigorous and realistic threat model that also provides the community with a useful metric for evaluations of class unlearning methods.

### B.11. Multiple Class Forgetting

| Forget Classes | $\text{ACC}_r \uparrow$ | $\text{ACC}_f \downarrow$ | MIA $\uparrow$ |
|---|---|---|---|
| 1 class | 77.08 (77.24) | 0 (77.03) | 100 (0.40) |
| 5 classes | 76.95 (77.10) | 0 (75.8) | 100 (0.96) |
| 10 classes | 74.84 (76.66) | 0 (77.04) | 99.42 (1.23) |

*Table 11.* **Multi-class forgetting performance on CIFAR-100 using ResNet18.** For each setting, we report the accuracy on the retained classes ($\text{ACC}_r \uparrow$), the accuracy on the forgotten classes ($\text{ACC}_f \downarrow$), and the MIA attack success rate (MIA $\uparrow$). Values in parentheses denote the corresponding metrics from the original (unforgotten) model. **Our method maintains high retained accuracy while fully forgetting target classes and preserving robustness under strong MIAs, demonstrating stable and scalable forgetting performance across varying numbers of classes.**

We evaluated our approach on multi-class forgetting tasks on CIFAR-100 using ResNet18, gradually increasing the number of randomly chosen forgotten classes from 1 to 10. Equation 7 computes a target distribution for all the samples in a forget class. Therefore, when unlearning multiple classes together, Equation 7 can be separately computed for the samples of each forget class. The only modification to this equation is that the probability mass will be redistributed to only the remaining classes, and the conditional probability for all the other forget classes will be forced to 0 as well. Our method achieves stable and effective forgetting across different numbers of target classes. As shown in Table 11, the retained accuracy ($\text{ACC}_r$) remains consistent with the original model, while the forgotten class accuracy ($\text{ACC}_f$) drops to zero in all cases. Additionally, the MIA attack success rate remains near-perfect, demonstrating that the unlearned model is indistinguishable from retrained baselines even under strong adversarial probing. These results indicate the robustness of our forgetting strategy in multi-class scenarios.

### B.12. Embeddings of Baselines

To better understand the geometric behavior underlying different unlearning methods, we visualize the embeddings of test samples using t-SNE, following the same setup as in Figure 3. In particular, we focus on the distribution of samples from the forgotten class (class 1) relative to the retained classes.

Figure 11 present the embedding visualizations for several baseline methods. Unlike the retrained model, which exhibits

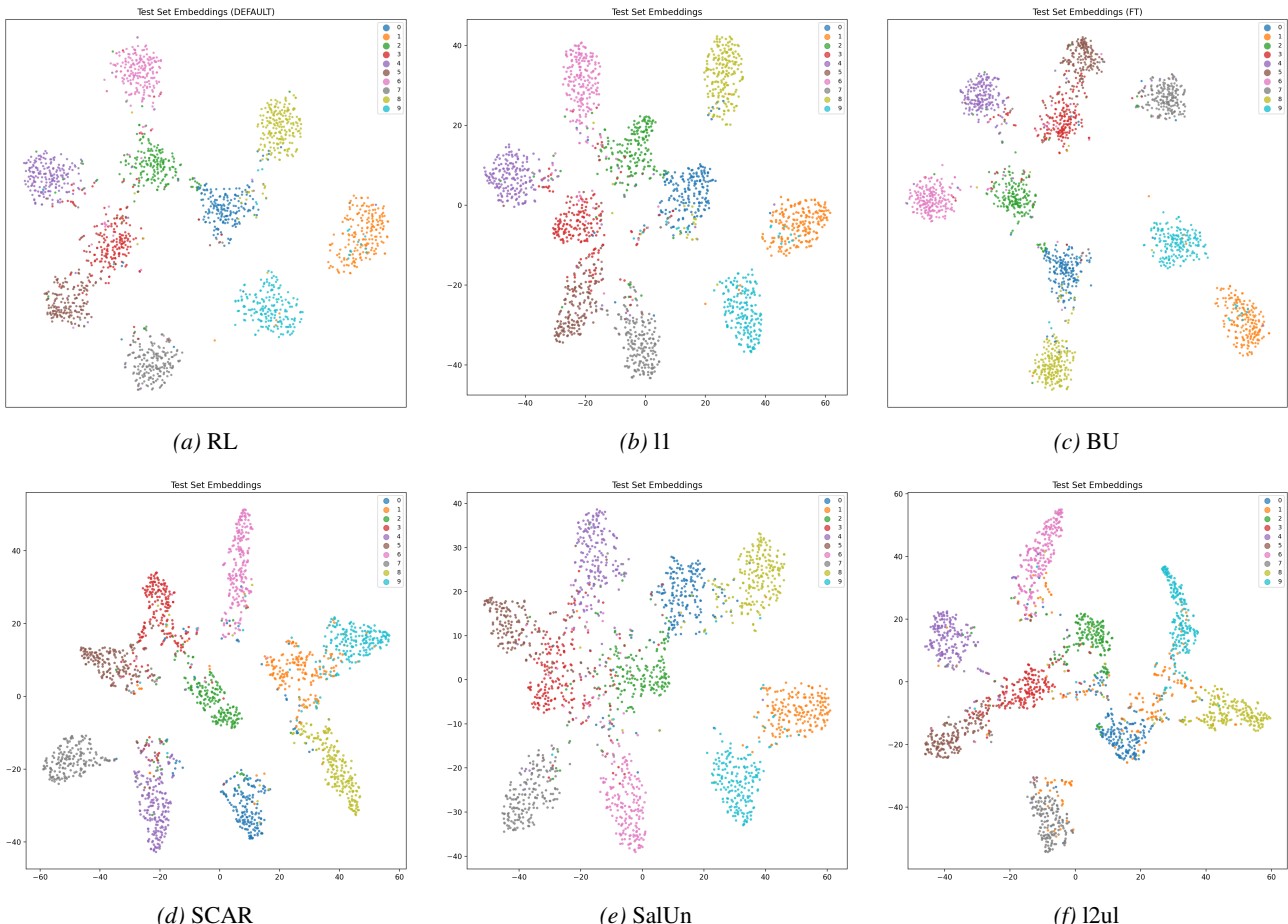

*(a)* RL       *(b)* ll       *(c)* BU

*(d)* SCAR       *(e)* SalUn       *(f)* l2ul

*Figure 11.* t-SNE embeddings of test samples for different unlearning methods on CIFAR-10. Each subplot shows the distribution of class representations after unlearning class 1. Unlike the retrained model, which exhibits a clear collapse of the forgotten class into a neighboring retained class, baseline methods either preserve a distinct cluster or exhibit unstructured dispersion.

a clear merging of the forgotten class into a semantically similar retained class (class 9), most baseline methods fail to reproduce this behavior.

We observe three distinct geometric patterns across baselines in Figure 11. First, methods such as *l1* and *SalUn* preserve a relatively well-separated cluster for the forgotten class, indicating that the model retains a distinct representation despite achieving low classification accuracy. This suggests that these methods primarily suppress decision boundaries without removing the underlying feature-level information.

Second, *l2ul* disperses the forgotten-class samples across multiple regions of the embedding space. However, this redistribution is largely unstructured and does not align with any specific retained class, deviating from the behavior of retrained models.

Finally, methods such as *UAM FT* and *SCAR* exhibit partial alignment, where a subset of forgotten-class samples shifts toward nearby retained classes. However, the collapse is incomplete and lacks the tight, concentrated merging observed in retrained models.

Overall, these observations highlight a key limitation of existing unlearning methods: while they may achieve low accuracy on the forgotten class, they fail to replicate the geometric transformation induced by retraining. In contrast, retrained models exhibit a structured collapse of forgotten-class representations toward semantically similar retained classes, resulting in low intra-class variance and consistent decision boundaries. This geometric discrepancy is precisely what CMIA is designed to capture.

### B.13. Sequential Multi-Class Unlearning

| Forget Classes | ACC$_r$ ↑ | ACC$_f$ ↓ | MIA ↑ | CMIA ↑ |
|---|---|---|---|---|
| 1 class | 77.09 | 0 | 100 | 96.29 |
| 5 classes | 74.11 | 0 | 100 | 92.31 |
| 10 classes | 72.62 | 0 | 99.45 | 87.07 |

*Table 12.* Sequential multi-class unlearning on CIFAR-100 with ResNet18.

We further evaluate whether our method can be extended beyond single-class forgetting to the setting of multiple forgotten classes. Rather than constructing a joint target distribution over all forgotten classes at once, we consider a simple *sequential forgetting* strategy, where classes are removed one by one. Specifically, given a sequence of target classes $f_1, \ldots, f_k$, we start from the original model and apply the same single-class TREW objective iteratively: after forgetting class $f_t$, the resulting model is used as initialization for forgetting class $f_{t+1}$.

We evaluated our approach on multi-class forgetting tasks on CIFAR-100 using ResNet18, considering settings with 1, 5, and 10 forgotten classes. For each setting, we randomly select the target classes to be forgotten, following the same protocol across all three cases. In particular, the 1-class setting corresponds to forgetting one randomly selected class, while the 5-class and 10-class settings correspond to forgetting randomly selected sets of 5 and 10 classes, respectively. As shown in Table 12, our method achieves stable and effective forgetting across different numbers of target classes. The retained accuracy (ACC$_r$) remains close to that of the original model, while the forgotten-class accuracy (ACC$_f$) drops to zero in all cases. Additionally, the MIA attack success rate remains near-perfect, indicating that the unlearned model stays difficult to distinguish from retrained baselines even under strong adversarial probing. These results demonstrate that our forgetting strategy remains robust as the number of forgotten classes increases.

Additionally, we evaluate privacy leakage using CMIA in the sequential multi-class setting. Since multiple classes are forgotten over time, we compute CMIA independently for each forgotten class and report the average across all forgotten classes. As shown in Table 12, CMIA remains high for small numbers of forgotten classes (96.29 for 1 class and 92.31 for 5 classes), indicating that the model closely matches the behavior of a retrained model and does not exhibit significant residual class-specific leakage.

However, as the number of forgotten classes increases to 10, CMIA drops to 87.07. This degradation suggests that sequential unlearning introduces accumulated approximation error, making it harder to faithfully reproduce the retrained distribution for all forgotten classes simultaneously. In particular, earlier forgotten classes may experience slight geometric drift as subsequent unlearning steps modify shared representations. Nevertheless, the CMIA score remains substantially higher

than typical baseline methods, indicating that our approach continues to preserve the desired geometric transformation and provides strong resistance to class-wise membership inference attacks even in more challenging multi-class settings.

## C. Future Work

Because of the differences in the nature of model training, objective, and definition of unlearning in these domains, LLMs, GNNs, and classification models have different approaches to unlearning. That being said, a few prior works in unlearning for classification models extend their approach to the setting of text-to-image generative models (Fan et al., 2023). It is interesting to note that our method can be extended to the setting of text-to-image generative models. Consider a generative model (e.g., diffusion model) that has learned the conditional distribution $p_\theta(x|c_f)$ for the forget class/concept $c_f$. To unlearn from this model, we would be able to set the tilted distribution as follows:

$$q^\star(x|c_f) \propto \sum_{c \in \mathcal{C}_r} \exp(\beta s(c, c_f)) \, p_\theta(x|c)$$

As our target distribution, where $s(c, c_f)$ measures semantic similarity (e.g., CLIP embeddings) between the forget concept $c_f$ and the remaining concepts $c \in \mathcal{C}_r$ and $\beta$ controls the tilt. Using this tilted distribution as the target distribution would bias the model toward concepts that are more semantically similar to $c_f$, leading to utility preservation of the model while unlearning class $c_f$. In general, our framework is very flexible and various similarity measures could be used to derive the desired form of the target distribution.

