# OpenReview forum: "Unlearning Isn’t Forgetting: Revealing Hidden Leakage in Class Unlearning Evaluations"
_ICML.cc/2026/Conference — ICML 2026 regular_

### Official Review · Reviewer_9sr1 · 2026-03-08

**Soundness:** 3
**Presentation:** 3
**Significance:** 2
**Originality:** 2
**Overall Recommendation:** 4
**Confidence:** 4

**Summary:**

This paper investigates a critical flaw in current class unlearning evaluation protocols: the failure to account for underlying class geometry and semantic similarity. The authors demonstrate that a model retrained from scratch (the "gold standard") naturally reassigns the probability mass of a forgotten class to its nearest semantic neighbors. Most approximate unlearning methods fail to replicate this behavior, making them vulnerable to a newly proposed CMIA. To mitigate this, the authors propose TREW, a fine-tuning objective that tilts the target model's output distribution to approximate the retrained model's bias toward similar classes.

**Compliance With Llm Reviewing Policy:**

Affirmed.

**Final Justification:**

The rebuttal addressed some minor points but did not fully resolve my primary concerns. As a result, my overall assessment remains unchanged. While the work shows potential, there is still substantial room for improvement in terms of technical soundness and generalizability.

**Key Questions For Authors:**

Please refer to Weaknesses.

**Limitations:**

yes

**Strengths And Weaknesses:**

**Strengths**

1. The paper identifies real and unexplored issue: existing evaluation metrics for class unlearning often only examine the probability assigned to the forgotten class, ignoring the broader structure of class predictions. This insight is both conceptually interesting and practically relevant for privacy evaluation.

2. The proposed CMIA is well-motivated and highlights a structural weakness in current evaluation protocols. By leveraging nearest-neighbor class probabilities, the attack reveals leakage patterns that standard MIAs fail to capture.


**Weaknesses**

1. While the paper compares against 10 established baselines, most of these are from 2020–2024. The authors should include very recent baselines from 2025 to strengthen the claim.

2. The "tilt" parameter β is fixed at 10 across various experiments. While the authors provide a theoretical range for β, the paper lacks a clear heuristic for practitioners to find the optimal β for new, unseen datasets without a retrained reference model.

3. The similarity score currently relies on cosine similarity of logit weights. While effective for CIFAR, its performance on datasets with thousands of classes (where "nearest neighbors" are less distinct) remains an open question.

4. You mention "neural collapse" as a justification for class-level constraints. Does the effectiveness of TREW diminish if the model has not reached the terminal phase of training (where neural collapse is less prevalent)?

5. Given that F(\beta) is convex, is it possible to implement a small-scale "proxy" retraining (on a subset of data) to automatically estimate the optimal tilt?

6. Although the paper provides propositions about KL divergence reduction, the theoretical results are relatively weak and rely on assumptions about similarity constraints. The connection between these assumptions and real neural networks is not deeply explored.

7. The work focuses entirely on class-wise forgetting in image classification tasks, which I believe limits the scope of the study. It would be valuable if the authors could extend their approach to generative models, such as large language models (LLMs).

---

> ### Author Rebuttal · Authors · 2026-03-31
>
> We thank the reviewer for the thoughtful and constructive feedback. We are glad that the reviewer found the core insight and the proposed CMIA to be conceptually meaningful and practically relevant. We address the concerns in the following:
>
> **W1 (more recent baseline)**.
>
> Following the suggestion by reviewer dKM7, we added a new baseline: Kim, H. (2025):
>
> https://tinyurl.com/y44b752h
>
> As the results show, it is prone to our newly developed CMIA and this further strengthens our claim
>
> **W2 (selecting β) + W5 (proxy retraining to estimate β).**
>
> In addition to the theoretical results mentioned by the reviewer, the ablations studies presented in table 7 and figure 7 show that TREW is robust to a wide range of positive β values, as long as extreme values are avoided. Also, performance is relatively stable across datasets and architectures. In practice, this suggests a simple heuristic for practitioners: selecting a moderate β (e.g., 5–15) is sufficient to achieve strong performance, without requiring access to a retrained reference model.
>
> Using a small proxy retraining is an interesting suggestion to approximate the target distribution and guide β selection in practice. However, one of our goals is to avoid retraining overhead altogether and derive a lightweight and effective method (as shown in Figure 6). We will add a discussion on this in the revised version of the paper.
>
> **W3 (cosine similarity)**.
>
> We agree that cosine similarity of logit weights may become less discriminative in settings with a large number of classes. We emphasize that TREW is agnostic to the choice of similarity function, and cosine similarity is only one instantiation. In the appendix (Table 8), we already explore alternative metrics (e.g., embedding-space distances), which show comparable trends. Also in section 6, we acknowledge the limitation of the similarity metric and lay it as a direction for future research. Still, we want to highlight the fact that in our experiments we also present results for datasets such as CIFAR100 and TinyImageNet with 100 classes where the original model cannot only achieve 60 to 70% accuracy and the nearest neighbors are less distinct.
>
> **W4 (neural collapse)**.
>
> We thank the reviewer for this insightful question. Neural collapse is used in our work primarily as an intuition to motivate the existence of structured class-level relationships, rather than as a strict requirement for TREW to be effective. TREW leverages relative similarity structure, which exists beyond the collapse regime. Empirically, we observe that even before terminal training, models exhibit non-uniform similarity patterns across classes. We will clarify this in the revised version.
>
> **W6 (theoretical results)**.
>
> We thank the reviewer for this important comment. Our theoretical contributions are intended to:
>
> Provide formal grounding for the tilting formulation via an information projection perspective
> Show that TREW improves alignment with retrained distributions under mild and interpretable conditions
>
> We agree that the theoretical guarantees are not strong in the sense of providing worst-case bounds for deep neural networks. Our goal is instead to offer a principled explanation of why tilting improves alignment with retrained models, rather than a fully certified unlearning guarantee.
>
> Importantly, the assumptions used in Proposition 3.2 are relatively mild. For example, the bounded variance assumption is natural in our setting, since similarity scores (e.g., cosine similarity) are inherently bounded. Similarly, the assumption that the retrained model exhibits bias toward semantically similar classes is consistently observed across datasets and architectures in our experiments.
>
> Despite these theoretical and empirical results, we want to emphasize that our work is focusing on proposing an **approximate unlearning method**. Most prior methods in approximate unlearning (e.g., Chen, M., et al. (CVPR 2023), Fan, C., et al. (ICLR 2024), Kodge, S., et al. (TMLR 2024), Bonato, J., et al. (ECCV 2024), Kim, H., et al. (NeurIPS 2025)) **do not provide any form of theoretical guarantees for unlearning**. Please refer to our response to Q1 of reviewer BB1z for further discussion on this line of work in comparison to certified unlearning literature.
>
> **W7 (other tasks)**.
>
> We agree that extending beyond image classification is an important direction. Our current focus is on image classification because: It provides a controlled setting to isolate class geometry effects and it aligns with prior class unlearning benchmarks in many prior research that focus on this task only (Chen, M., (2032), Kodge et al. (2024), Bonato et al. (2024), Panda, S., (2025), Wang, J., (2026)), etc.).
>
> However, the core idea of TREW is general and can extend to other domains. In Appendix D, we have proposed one such extension to unlearning in diffusion models for further exploration in future work.
>
> **References:** https://tinyurl.com/56x2tysc

---

> > ### Author Rebuttal · Reviewer_9sr1 · 2026-04-03
> >
> > I really appreciate the authors’ detailed reply, but my concern still lies in the fact that the research scenario is too narrow.
> >
> > First, can the authors provide the full paper titles for the works cited in your response? Since you only mention author's first name, it is difficult for me to search for any specific information regarding these papers.
> >
> > Furthermore, the year cited for '(Chen, M., (2032)' appears to be a typographical error that needs correction. It is also critical to provide the paper titles for the recent works you mentioned, such as Panda, S. (2025) and Wang, J. (2026), which you claim also only consider a single forgetting scenario in image classification. I would like to point out that at least one of the works you cited, Bonato et al. (2024), is not limited to class-wise forgetting alone. My concern is not necessarily that only focusing on image classification tasks is unacceptable, but rather to emphasize that you need to provide a broader range of scenarios within image classification to demonstrate the generalizability of your method.
> >
> > Regarding the extension of the current work to other tasks, could the authors explain whether the random data forgetting scenario can be addressed using TREW? Specifically, is it possible to calculate instance-wise similarity to derive the projection and bias targets in the retain set, similar to how you currently handle class-level similarities?
> >
> > While I have read your discussion on extending this to diffusion models, I believe the work would be significantly more solid if you were to implement it.
> >
> > If you insist on staying within the image classification domain, then subclass forgetting is a scenario that is certainly worth discussing. Observing TREW in a subclass context—where you must forget specific data points within a class while simultaneously preserving the performance of the remaining data points in that same class—would be highly meaningful. This setup would provide a clearer observation of how TREW effectively adjusts decision boundaries.

---

> > > ### Author Response · Authors · 2026-04-05
> > >
> > > We would like to thank the reviewer for evaluating our responses and engaging in a constructive discussion.
> > > We would like to clarify that all references were provided in the “References” link at the bottom of our response. For your convenience, we include the link again here:
> > >
> > > https://tinyurl.com/56x2tysc
> > >
> > > Next, we would like to emphasize that random-sample unlearning is a different setting from class unlearning, which is the main focus of our paper. In class unlearning, removing an entire class leads to a structured redistribution of probability mass toward semantically similar retained classes, which is precisely the phenomenon that TREW is designed to model. By contrast, in random-sample unlearning, the class itself remains present in the training distribution, so the corresponding decision region is generally preserved. This distinction between local (sample-level) and global (class-level) unlearning has also been noted in some prior work that only considered the class unlearning setting: Chen, M. (2023), Kodge, S. (2024), Panda, S. (2025), Wang, J. (2026).
> > >
> > > Extending TREW to the setting of random sample unlearning would likely require modified instance-conditioned or sample-dependent similarity targets rather than the class-level similarity used here. We therefore view random-sample unlearning as a promising extension of the framework, but not as a setting addressed by the current paper.
> > >
> > > Motivated by the reviewer’s suggestion, we also evaluated TREW in a finer-grained subclass forgetting setting within image classification, which better aligns with the scenario that TREW targets. Our results in this setting further clarify the advantages of our method. CIFAR-100 contains 100 classes grouped into 20 superclasses (e.g., aquatic mammals, fish), each consisting of five fine-grained classes. In this experiment, we train a 20-class classifier on the superclasses and then consider unlearning a fine-grained class (i.e., a subclass of a broader class). The following results show the performance of the unlearning method when removing the beaver class, which is a subclass of the “aquatic mammals” superclass. The remaining subclasses within the aquatic mammals superclass should still be correctly predicted as aquatic mammals, while the beaver class should be unlearned. As the results show, TREW outperforms other recent methods and produces a model that is closer to the retrained model. We will include this new experiment in the revised version of our paper.
> > >
> > > Method        | ACCf (test) | ACCr (test) | MIA    | Average Gap (%)
> > > ------------- | ----------- | ----------- | ------ | ----------------
> > > Retrain       | 21.00%      | 85.57%      | 60.17% | -
> > > FT            | 43.00%      | 80.61%      | 84.83% | 17.08
> > > SCAR          | 25.57%      | 82.39%      | 78.12% | 8.88
> > > UAM           | 24.38%      | 81.43%      | 76.17% | 7.84
> > > TREW (ours)   | 22.43%      | 82.75%      | 65.67% | 3.25
> > >
> > >
> > > We agree with the reviewer that implementing our idea for diffusion models would strengthen the effectiveness of TREW. However, each of these areas has become increasingly specialized and requires its own tailored treatment and set of experiments to properly justify effectiveness.
> > >
> > > Moreover, we would like to emphasize that the paper contributes both a new unlearning objective (TREW) and **a new evaluation method (CMIA), which exposes a failure mode that is missed by existing class-unlearning evaluations**. Our goal in this submission is therefore not to maximize task breadth, but to study this failure mode carefully and show that TREW mitigates it in the target setting.
> > >
> > > Lastly, we would like to thank the reviewer again for the constructive discussion, which has helped improve our paper. If the reviewer finds our responses and new experiments satisfactory, we would greatly appreciate their consideration in increasing their score to support our work and its contributions to the community.

---

### Official Review · Reviewer_wL4F · 2026-03-12

**Soundness:** 3
**Presentation:** 2
**Significance:** 2
**Originality:** 3
**Overall Recommendation:** 4
**Confidence:** 4

**Summary:**

The paper proposes a class membership inference attack that probes whether a model assigns probabilities to neighboring classes to detect unlearned classes. This is based on the observation that models retrained from scratch assign the samples from the unlearned class to the nearest class with the most similar features. Models unlearned using existing techniques do not do this.
As a solution, the paper proposes Tilted Reweighting, which is an additional loss signal when fine-tuning the model on the forget set, which reweights the probabilities towards classes with high similarity to the forget class. The experimental results demonstrate that this method outperforms existing unlearning techniques.

**Compliance With Llm Reviewing Policy:**

Affirmed.

**Final Justification:**

All my concerns have been addressed and I adjusted my score.

**Key Questions For Authors:**

Q1: In what scenario would someone wanna unlearn a whole class? Why would it be a problem if someone would know that this class ever existed in this scenario?
Q2: Wouldn't the insight from 3.2 change if the class that was unlearned is removed from the final layer and then just fine-tune the model on the samples of the forget set to match with the closest class? Wouldn't this be a natural change to the model if someone unlearns a whole class?

**Limitations:**

Yes

**Strengths And Weaknesses:**

Strengths:
- the insight that models behave differently when classes are removed instead of retrained makes sense and seems to be novel
- the theoretical insights are interesting and novel

Weaknesses:
- I am not sure about the scenarios where this would be useful. Are there any scenarios where removing a whole class from a model would be needed? (also see my questions down below)
- There are a lot of assumptions for the CMIA attack.
- It is a bit hard to follow to understand how CMIA works. For example it is not clear to me how the retrained models for the CMIA are trained and what the forget class $D_{f-test}$ has to do with splitting the test set of each class $r_i$.
- In Table 1 it is not clear which class was unlearned now for these datasets. Was there only one class unlearned? Or were there multiple different classes unlearned in multiple runs for which then the average and the standard deviation was given?
- In the proof for proposition 3.1 shouldn't it be a negative sign for the exponent?:
  $q_\beta(y) = \frac{\tilde{p}(y) e^{-\beta s_y}}{\sum_j \tilde{p}(j) e^{-\beta s_j}}$

Misc:
- The notation for the dataset in 3.1 doesn't make sense with the classes K. This would mean that N=K, if $y_f$ is the class that should be unlearned
- The values in Table 1 are way to small. Even in a pdf it is hard to read them

---

> ### Author Rebuttal · Authors · 2026-03-31
>
> We thank the reviewer for the insightful feedback and for recognizing the novelty of our class-geometry perspective and theoretical analysis. Here are our responses to their concerns.
>
> **Q1 + W1 (importance of unlearning a class)**.
>
> We thank the reviewer for this important question. In privacy and regulatory settings (e.g., “right to be forgotten”), requests may apply to categories of data rather than individual samples. In many real-world settings, revealing the presence of a class in the training data can itself be sensitive. For example, in medical applications, if a model can reveal that it was trained on data corresponding to a particular condition (e.g., a rare disease), this may disclose information about the underlying dataset or participating patients, even if the model no longer predicts that class. Similarly, in privacy-sensitive domains, the presence of a protected attribute (e.g., a demographic group) in training data may need to be concealed, not just suppressed at inference time.
> Finally, class unlearning has emerged as a distinct research direction, reflecting the fact that removing a class or concept differs fundamentally from instance-level deletion. Therefore, a separate line of research has focused explicitly on class unlearning (Chen, M., (2032), Kodge et al. (2024), Bonato et al. (2024), Panda, S., (2025), Wang, J., (2026)).
>
>
> **Q2 (removing the forget class)**.
>
> We thank the reviewer for this insightful question. Note that, if as suggested by the reviewer, we merely change the target label for the forget samples to the most similar class, that gives us the extreme case of equation (7) when $\beta$ goes to infinity (as explained in line 237). So the formulation of equation (7) contains the reviewer’s solution as a specific case. Also, as Figure 5 in the paper shows, for some classes, such as Frog in CIFAR10, there is not a single class that can be used as the replacing label, but the distribution might  be split among several similar classes..
>
> **W2 (assumptions for CMIA)**.
>
> CMIA does not rely on additional assumptions beyond standard MIAs. The only assumptions we make is access to the $D-D_f$ to train the shadow models. We even drop this assumption and only assume access to the publicly available samples in our black-box setting presented in section C.10 to show it still works. Note that our threat model is a standard one based on SOTA MIA for machine unlearning (Hayes et al. 2025) and the procedure of using the training samples or public samples to train shadow models is the common practice in MIA literature (Carlini et al. (2022) and Zarifzade et al. (2024)).
>
> **W3 (splitting the test set in CMIA)**.
>
> Thank you for the feedback. We will add additional schematic visualizations to further clarify how CMIA works. As the figure in https://tinyurl.com/4bcm587a shows, for each class $r_i$ we split the test samples into three groups: 1) test samples with label $r_i$; 2) test samples with label $y_f$; and 3) test samples with any label other than $r_i$ and $y_f$.  Using retrained models, we train a binary classifier on the logit values for label $r_i$​ to distinguish (1) vs. (3).
>
> We then apply this classifier to forget-class samples (group 2). If many of these samples are classified as (1), it indicates that the forget class is mapped toward $r_i$​, i.e., $r_i$​ is a nearest neighbor of the forgotten class in logit space. CMIA selects the class with the strongest such signal and evaluates whether the unlearned model exhibits the same behavior as the retrained ones.
>
>
>
> **W4 (table 1)**.
>
> Table 1 reports results for a single forget-class averaged over multiple runs. The notation in the “Dataset” column specifies this setting: for example, (8 → 3) means class 8 is unlearned, and class 3 is identified by CMIA as its nearest neighbor.
>
> In contrast, Table 2 (and Tables 4–5 in the appendix) report averaged results over multiple forget classes (10 randomly selected classes per dataset). We will make this distinction explicit in the text and captions to avoid confusion.
>
>
> **W5 (proposition 3.1)**.
>
> We appreciate the careful reading. The sign in Equation (7) is intentional: positive β tilts mass toward higher-similarity classes. However, in the proof of Proposition 3.1, the Lagrange multiplier should be reparameterized explicitly before moving from $q_β(y)\propto \tilde p(y)e^{−β s_y}$ to the final $e^{+βs_y}$ form. We will clarify this in the revision. However, it is just a reparametrization and does not affect the result of the proof.
>
> **W6 (notations)**.
>
> Here, $N$ denotes the number of data samples and $K$ denotes the number of classes, which are independent quantities. The dataset is defined as $\mathcal{D} = {(x_i, y_i)}_{i=1}^N$ with labels $y_i \in \mathcal{Y}$, where $|\mathcal{Y}| = K$. The forget class $y_f \in \mathcal{Y}$ is a single class index, and $\mathcal{D}_f$ consists of all samples in $\mathcal{D}$  with label $y_f$.
>
> **References:** https://tinyurl.com/56x2tysc

---

> > ### Author Rebuttal · Reviewer_wL4F · 2026-04-01
> >
> > Thank you for the clarification. All my concerns have been addressed and I will adjust my score.

---

> > > ### Author Response · Authors · 2026-04-04
> > >
> > > Thank you very much for your constructive feedback and comments. We are glad that our responses have fully addressed your concerns and strengthened the paper.

---

### Official Review · Reviewer_dKM7 · 2026-03-13

**Soundness:** 3
**Presentation:** 2
**Significance:** 1
**Originality:** 2
**Overall Recommendation:** 4
**Confidence:** 4

**Summary:**

**[Important] This paper does not appear to follow the ICML formatting guidelines. The font and margins differ from those of other submissions. The evaluation below is provided under the conditional assumption that this issue will be resolved.**

This paper is based on the observation that class geometry can cause information leakage about the forgotten class, and proposes a method that leverages neighboring classes to detect unlearned samples. Based on this observation, the authors introduce a new unlearning method called TREW. While the overall idea is interesting, the experimental setup is limited to overly simple datasets and relies on relatively old baselines, which weakens the empirical support for the claims. Addressing these aspects would strengthen the paper.

**Compliance With Llm Reviewing Policy:**

Affirmed.

**Final Justification:**

The authors have acknowledged the formatting inconsistency. While this is a requirement that all papers should adhere to, it was not followed in this case; therefore, I will leave the judgment on this matter to the AC.

That said, the authors’ responses have satisfactorily addressed all of my concerns. In particular, the additional experiments, including comparisons with recent methods, provide stronger evidence of the effectiveness of the proposed approach in the unlearning setting.

Accordingly, I have updated my evaluation and increased my score from 3 to 4. Thank you for the clarifications and efforts.

**Key Questions For Authors:**

Refer to Strengths And Weaknesses

**Limitations:**

N/A. Authors already noted the limitation in their paper.

**Strengths And Weaknesses:**

### Soundness

Strength.

- The observation that forgotten images are consistently classified as other images is an interesting finding.

Weakness.

N/A

### Presentation

Strength.

- Figure 1 is very well presented and allows the reader to quickly understand the core idea.

Weakness.

- Figure 1 should more clearly demonstrate that the phenomenon is general. Although Figures 4 and 5 in the Appendix attempt to support this, they are somewhat difficult to interpret. It may be helpful to present the results through repeated experiments and summarize them using ratios.
- It may be beneficial to explain the CMIA phenomenon in more detail and then introduce TREW in a separate section. Currently, CMIA and TREW must be read together, which somewhat dilutes the impact of each component.

### Significance

Strength.

- N/A

Weakness.

- In Figure 3, it would be more meaningful to also show how other models behave, rather than only presenting the authors' model.
- In the experiments, the following recent methods should be considered. At least the methods listed below should be included, as the current baselines are largely outdated (most are from before 2024).

    - UAM (Unlearning-Aware Minimization, NeurIPS 2025)
    - Lagrangian and Wasserstein-based method (Machine Unlearning under Retain–Forget Entanglement, ICLR 2026)

    Although the second method may be too recent to include, the first one should reasonably be compared. In addition, these papers should be added to the Related Work section. I recommend updating Tables 1 and 2 accordingly.

- Overall, the experimental design appears somewhat limited. Evaluation on ImageNet or with a ViT-Base model would strengthen the empirical validation.

### Originality

Strength.

- The discovery of CMIA is clearly a novel contribution. The finding that forgotten classes tend to move to other classes is important, and the ability to identify which class has been forgotten using logits is also meaningful.

Weakness.

- However, the proposed method TREW is somewhat less convincing. While identifying CMIA is valuable, the method is designed specifically to optimize this measure, which may naturally lead to favorable results. In practice, MIA performance is worse than SVD, suggesting that forgetting may not be fully achieved. In addition, the reported performance of L1 sparse and Salun appears somewhat questionable, as prior and recent papers seem to report higher performance for these methods than what is shown in this paper.

---

> ### Author Rebuttal · Authors · 2026-03-30
>
> We thank the reviewer for their careful review and the constructive feedback. We are glad that the reviewer recognized the novelty of our observation (CMIA) and the clarity of figures and results. Below we address the concerns in detail.
>
> **W1 (formatting concern).**
> We thank the reviewer for pointing this out. We identified that the slight formatting issue was due to inconsistency with an imported package. We have fixed this for the revised version.
>
> **W2 (generalizing Figure 1).**
> We are glad that the reviewer found Figure 1 effective in conveying the core intuition. To show its generality to other classes and summarizing them to one figure, we have created this new plot for CIFAR10, where each class (each row) shows the predictions of a model that is retrained from scratch without that class:
>
> https://tinyurl.com/yf4tjybz
>
> As the results show a similar trend exists for all the other classes with predictions being biased to a few of the remaining classes.
>
> We will also revise the presentation to more clearly separate (i) the empirical observation and attack (CMIA), and (ii) the mitigation strategy (TREW). In particular, we will restructure the paper to highlight CMIA as a standalone finding that motivates the design of TREW.
>
> **W3 (recreating Figure 3 for other methods).**
> We have created similar figures for some of the other more recent baselines which further demonstrates the mismatch of the unlearned models with the retrained ones:
>
> https://tinyurl.com/yckdcphz
>
> We will add these figures to the appendix of our paper to further improve the results.
>
> **W4 (new baselines).**
> We appreciate the suggestion about these newer methods in machine unlearning. We will use both works to strengthen our related work section. We also included UAM in our experiments:
>
> https://tinyurl.com/y44b752h.
>
> As the results show, its CMIA score is 66.71%, which is substantially below our method’s 95.82%. This suggests that, although UAM performs well by conventional metrics, it remains vulnerable under CMIA, and this further strengthens our claim on the importance of including this new evaluation method to the machine unlearning literature.
>
> **W5 (SVD, L1 sparse, and Salun)**.
>
> Although TREW is motivated by our observations from CMIA, it is grounded in a principled approach and theoretical formulation to achieve robustness against CMIA. We would like to emphasize that we have evaluated TREW not only against CMIA, but also using existing evaluation metrics (Table 2) and SOTA MIA for machine unlearning.
>
> Please note that the MIA result for SVD ($97.20$) is comparable to (and not better than) TREW ($97.65$) in Table 2. While the MIA used in Table 2 is useful for consistency with prior work, we additionally provide CMIA and evaluate using the much stronger U-LiRA attack (Hayes et al., 2025), which more clearly highlights the performance gap.
>
> Also, we would like to emphasize that we used the official implementations of each method from their repositories and ran all experiments using their suggested settings. The differences observed for Salun and L1-sparse methods might stem primarily from the MIA used for evaluation. The MIA used in Table 2 follows Kodge et al. (2024) and is specifically designed for class unlearning evaluation. Additional differences may arise from other factors, such as averaging over 10 classes in our experiments, whereas this setting is not clearly specified in some prior work, such as Salun.
>
>
>
> **W6 (other datasets)**.
>
> We would like to emphasize that the models and datasets used in our paper are standard choices across the recent class unlearning baselines. While our paper presents results on ResNet-18 and VGG for MNIST, CIFAR10, CIFAR100, and TinyImageNet, the experiments in recent baselines (Bonato, J., et al. (2024), Kodge, S., et al  (2024), Cha, S., et al (2024)), and also the most recent work mentioned by the reviewer (Cheng, J., et al (2026)) use **the same models and datasets (or a subset of them)** for their experiments. Nevertheless, we acknowledge that a broader evaluation would further strengthen the work.
>
> We hope that we have adequately addressed the reviewer’s concerns and that they will consider raising their score in support of our work and its contributions to the machine unlearning community.
>
> **References:** https://tinyurl.com/56x2tysc

---

> > ### Author Rebuttal · Reviewer_dKM7 · 2026-04-03
> >
> > The authors have acknowledged the formatting inconsistency. While this is a requirement that all papers should adhere to, it was not followed in this case; therefore, I will leave the judgment on this matter to the AC.
> >
> > That said, the authors’ responses have satisfactorily addressed all of my concerns. In particular, the additional experiments, including comparisons with recent methods, provide stronger evidence of the effectiveness of the proposed approach in the unlearning setting.
> >
> > Accordingly, I have updated my evaluation and increased my score from 3 to 4. Thank you for the clarifications and efforts.

---

> > > ### Author Response · Authors · 2026-04-04
> > >
> > > Thank you very much for your thoughtful and constructive feedback throughout the review process. We are glad that the additional experiments and clarifications have addressed your concerns and strengthened the paper.
> > >
> > > Regarding the formatting point, we have already corrected this minor inconsistency, which was subtle enough to go unnoticed on our end.
> > >
> > > Thank you again for your support and for increasing your score; we do appreciate it.

---

### Official Review · Reviewer_W7Ef · 2026-03-18

**Soundness:** 3
**Presentation:** 4
**Significance:** 3
**Originality:** 3
**Overall Recommendation:** 4
**Confidence:** 3

**Summary:**

This work studies class unlearning, where the goal is to remove the semantic concept of an entire class from a model, rather than just forgetting a few individual training examples. The authors argue that this setting is fundamentally harder than instance-wise unlearning, because even after removing samples from the forget class, the model can still retain class-specific patterns and relationships. To expose this problem, they introduce CMIA, a new membership inference attack that detects the presence of a forgotten class by leveraging its nearest-neighbor classes. Their experiments show that many previous unlearning methods remain vulnerable to this attack, largely because they were designed for instance-wise unlearning and because they only aimed to reduce accuracy or increase loss on forget samples, which is not sufficient for true class removal. In response, the authors propose a new fine-tuning loss with only light computational overhead, designed to better approximate the behavior of a model retrained from scratch without the forget class, especially in terms of the retain-class distribution. Experiments show that this approach is more robust than prior state-of-the-art methods, both against standard MIAs and against CMIA.

**Compliance With Llm Reviewing Policy:**

Affirmed.

**Key Questions For Authors:**

No questions.

**Limitations:**

Yes.

**Strengths And Weaknesses:**

*Soundness*:
The paper generally does a good job of supporting its claims through both theory and experiments. On the theoretical side, the two proposed proofs appear sound and rely on reasonable assumptions. On the experimental side, the results are useful for building intuition, but the considered datasets seem relatively simple and structured. It would therefore be valuable to test the method on more challenging settings, where class boundaries are less clearly separated and per-class accuracy is not naturally very high. In that sense, the claim around line 125, namely that “this behavior is not specific to model architecture but emerges from underlying data-level class similarities,” seems to require stronger empirical support, ideally through a broader range of architectures and datasets. There is also a point in Table 2 that may need clarification: if the average gap is computed across ACCr, ACCf, MIA, and CMIA relative to the Retrain baseline, then for forget-set accuracy the gap should arguably be defined as "Retrain minus Unlearning", since lower values are better there. Using the opposite direction may understate the degradation and make the average gap less representative; for readability, a normalized version of the gap would also be preferable. Finally, the ablation study in Section C.7 does not show variability, which makes the conclusions harder to assess. If the experiments were run only once, then claims such as selecting the best $\beta_f$ are not fully supported by the presented results. For example, the higher retain accuracy reported for $\beta=5$ compared with $\beta=0$ could just as plausibly be due to run-to-run randomness as to a meaningful underlying trend.

*Significance*:
The paper advances the field of applied unlearning and, although the setting is domain-specific, the proposed ideas may offer useful improvements for other existing methods as well as for future unlearning approaches. At the same time, there remains some doubt about how well the method would perform on more complex models and datasets. This should be investigated further, and one possible relevant direction could be topic removal in LLMs.

*Presentation*:
The paper is written in a coherent and well-structured manner. It clearly introduces the problem and explains its importance, positions the work with respect to the related literature, and identifies the key challenges. It also builds the necessary intuition behind these challenges, which helps motivate both the proposed attack and the unlearning method. In addition, the tables and figures present the results clearly and in an easy-to-understand way. As a side note, one should avoid making overly strong claims related to previous literature, unless they are absolutely sure (e.g., final sentence of the conclusion with "outperforming ALL existing methods" and "other existing metrics").

*Originality*:
The work highlights the limitations of previous MIA evaluation methods and provides new insights into when and why they fail, especially in the class-unlearning setting. It proposes a new attack that exposes these weaknesses through a relatively simple modification of an existing method. It also introduces a new unlearning method that is robust to both standard and class-aware MIA attacks. In addition, the TREW method appears to be relatively easy to integrate into existing fine-tuning-based unlearning approaches.

---

> ### Author Rebuttal · Authors · 2026-03-30
>
> We thank the reviewer for the thoughtful and constructive feedback. We are encouraged that the reviewer finds the paper sound, well-motivated, and clearly presented. We also appreciate the recognition of the novelty of our work in introducing a new attack that reveals shortcomings of existing methods. We address the raised concerns below.
>
> **Weaknesses (More datasets and models).**
>
> We would like to emphasize that the models and datasets used in our paper are standard choices across the recent class unlearning baselines. While our paper presents results on ResNet-18 and VGG for MNIST, CIFAR10, CIFAR100, and TinyImageNet, the experiments in recent baselines (Bonato, J., et al. (2024), Kodge, S., et al  (2024), Cha, S., et al (2024)), and the new baselines mentioned by reviewer dKM7 (Cheng, J., et al (2026)) use **the same models and datasets (or a subset of them)** for their experiments. We also want to emphasize that in addition to the outdated regular MIA used in most prior work, we used **a stronger recent MIA (Hayes, J., et al, (2025)) and our newly developed CMIA** for a more comprehensive evaluation.
>
> While we agree with the reviewer, that more datasets and more models would enhance the results, we believe that some of the current datasets (CIFAR100 and TinyImageNet) match some of the properties the reviewer is looking for: “class boundaries are less clearly separated and per-class accuracy is not naturally very high” as the accuracy of the original models on these are much lower (e.g., $63.49$ for TinyImageNet).
>
> We also added one more recent baseline, UAM (Kim, H., et al., 2025), as requested by reviewer dKM7, which still shows the same shortcoming against our proposed CMIA: https://tinyurl.com/y44b752h
>
> It is noteworthy that the core idea of TREW is general and can be extended to other domains. In Appendix D, we propose one such extension to unlearning in diffusion models as a direction for future work.
>
> **Weaknesses (The average gap computation).**
>
> We thank the reviewer for catching this. The good news is that this problem only applies to the average gap computed to one of the baseline methods (GA) in table 2, which does not achieve $ACC_f$ of $0$. We will update all the tables accordingly in the revised version. Importantly, this correction does not affect the main conclusions of the paper: our method still achieves the best overall performance and improves over the strongest baseline UAM by $29.11$ absolute points in CMIA.
>
> **Weaknesses (Variability over multiple runs for the results in appendix C.7).**
>
> We want to emphasize that the presented results in Table 8 are averaged over 3 different runs. However, we agree with the importance of including the standard deviations and we will include them in the revised version of the paper. Still we want to highlight that **we did not focus on negligible differences in the remaining accuracy** for $\beta_f$ in table 7, but by improvement in the unlearning **we considered the noticeable differences in the CMIA scores**. As mentioned in line 947: “the model preserves strong predictive performance on the retained data” (we considered those values almost the same), and in line 948 we pointed out that “unlearning effectiveness improves significantly” because “the CMIA score rises to $82.1$” which is a noticeable improvement from $47.2$ in $\beta_f=0$ and $76.1$ in $\beta_f=5$, and cannot be contributed to randomness of training (especially considering that results are averaged over 3 runs).
>
>
> **Weaknesses (Overly strong claims related to previous literature).**
>
> We will revise overly strong claims, such as “outperforming all existing methods” and limit our claims to the included baselines.
>
> We again thank the reviewer for the constructive comments and recognizing our contributions. We hope the reviewer raises their score to indicate a clearer signal to the AC about our work and its potential contributions to the machine unlearning community.
>
> **References:** https://tinyurl.com/56x2tysc

---

> > ### Author Rebuttal · Reviewer_W7Ef · 2026-04-03
> >
> > The authors’ revisions largely address my concerns. However, I do not suggest any score adjustment, as my initial scores were already given with some allowance for these points.

---

> > > ### Author Response · Authors · 2026-04-04
> > >
> > > Thank you for your constructive feedback and for reviewing our responses.

---

### Official Review · Reviewer_BB1z · 2026-03-23

**Soundness:** 2
**Presentation:** 3
**Significance:** 3
**Originality:** 3
**Overall Recommendation:** 4
**Confidence:** 3

**Summary:**

The paper identifies a critical issue in how the privacy of class unlearning methods is currently evaluated. The authors argue that merely forcing a model to assign zero or low probability to a "forget class" is superficial. When a model is genuinely retrained from scratch without a specific class, it systematically redistributes its predictions for those forgotten samples toward semantically similar, retained classes. Since most existing approximate unlearning methods fail to replicate this underlying geometric shift, they leave residual structural signals that can be exploited by adversaries.To address this, the authors introduce a novel Class Membership Inference Attack (CMIA) to expose this leakage , and a new fine-tuning objective called Tilted REWeighting (TREW) to mitigate it.

**Compliance With Llm Reviewing Policy:**

Affirmed.

**Key Questions For Authors:**

1. In Introduction, the author claims that efficient exact unlearning exists for convex optimization. Can you point out which work achieved this and how?

2. How to generalize the one-class forgetting to multi-class forgetting. Does a sequential manner always work?

3. see above: can the authors justify the why (7) is a good reweighing model, capturing the behavior of the objective perfectly-unlearned model. Although the author demonstrate that mathematically this is essentially a projection onto the $l_1$ convex hall, it would be better to directly present the approximation error of (7) and the prediction vector from the ground-truth perfectly-unlearned model.

**Limitations:**

yes

**Strengths And Weaknesses:**

Strengths:
+ The prediction on the prediction behavior of the perfectly concept/class-unlearning by reassigning the confidence weight proportionally to the similarity between the target class and remaining class. This methodology is interesting.

+ Extensive experiments have been provided to demonstrate the effectiveness of proposed method.

Weakness:
- Despite the novelty of the methodology, my major technical concern is from the currently-proposed solution by directly reassign the confidence score promotional to the cosine similarity. Qualitatively the proportional confidence score increase make senses, but quantitatively I think more work is needed to justify why (7) is the right model to use. In particular, I am wondering whether a more model-dependent reweighing model can be lightly learned from the given model, to be unlearned, that can more tightly mimic the objective perfectly-unlearned model.

---

> ### Author Rebuttal · Authors · 2026-03-31
>
> We thank the reviewer for the thoughtful and constructive feedback. We are glad that you found the problem formulation, CMIA attack, and overall methodology interesting and well-supported experimentally. Below we address your concerns in detail.
>
> **Weaknesses and Q1 (Eq. (7)):** We want to emphasize that our choice of Eq. (7) is not heuristic; it arises as the information projection of the original model distribution onto a constrained space. We basically enforce only the following:
>   - Removal of the forget class probability mass
>   - A first-moment constraint over class similarity
>
> As shown in Prop. 3.1, among all distributions satisfying these constraints, Eq. (7) is the **closest to the original model in KL divergence**. This yields a **maximum-entropy/minimum-distortion solution**, which is a principled choice for the model when:
> - The true retrained distribution is unknown
> - We want minimal intervention to keep model's utility on the remaining samples
>
> Additionally, our theoretical result (Prop. 3.2) shows that for a wide range of β values, the tilted distribution reduces KL divergence to the retrained model compared to naive renormalization. Despite these theoretical and empirical results, we want to emphasize that our work is focusing on proposing an **approximate unlearning method**. Most prior methods in approximate unlearning (e.g., Chen, M., et al. (CVPR 2023), Fan, C., et al. (ICLR 2024), Kodge, S., et al. (TMLR 2024), Bonato, J., et al. (ECCV 2024), Kim, H., et al. (NeurIPS 2025)) **do not provide any form of theoretical guarantees for unlearning**. On the other hand prior work on certified unlearning, which are accompanied by theoretical guarantees on the approximation error, make various assumptions that do not hold for deep learning models:
>
> - Sekhari, A. et al. (NeurIPS 2021), assume that the model is strongly convex (assumption 1 in Section 4).
> - Chien, E., (ICLR 2023) make a similar assumption to bound the inverse Hessian of the loss with respect to the parameters (proof of Theorem 4.3 in Appendix A.7).
> - Zhang, B., et al. (ICML 24) uses similar assumptions on bounding the inverse Hessian matrix, but they exploit the local convex approximation (Lemma 3.3 in Section 3), and rely on Lipschitz continuity of Hessian of the loss (Assumption 3.2 in Section 3).
>
> Reviewer’s suggestion on a more model-dependent reweighing model that can be lightly learned from the original model, is an interesting future direction; still one should consider the computation complexity that these solutions might entail. It also requires a new model for each unlearning request, which makes it computationally prohibitive soon. Finally, while our proposed method might not be the ultimate solution, it provides a strong, lightweight, and principled baseline for future work.
>
>
> **Q1 (certified unlearning for convex models):**  Thank you for pointing this out—we will elaborate on this in the revision. Examples of exact or certified unlearning in convex settings include:
>
> Guo et al., 2020 propose efficient certified unlearning method for **l_2** regularized linear models (e.g., logistic regression). Izzo et al., 2021 also focus on certified unlearning for linear and logistic models. Sekhari, A. et al. (NeurIPS 2021) assume strong convexity. More recent methods, try to expand the scope of certified unlearning literature (Zhang, B., et al. (ICML 24)) by making other assumptions (Lipschitz continuity of Hessian of the loss). We will update the related work section to further clarify the distinction of approximate unlearning from this line of work.
>
>
> **Q2 (multi-class unlearning):**
>
> Eq. 7, computes a target distribution for all the samples in a forget class. Therefore, when unlearning multiple classes together, Eq. 7 can be separately computed for the samples of each forget class. The only modification to Eq. 7, is that the probability mass will be redistributed to only the remaining classes, and the conditional probability for all the other forget classes will be forced to 0 as well. In the multi-class unlearning experiment in section C.11 of our paper we have the results of this procedure. As the results show, increasing the number of classes only leads to a slight degradation of the mode’s utility.
>
> If the unlearning request for different classes is received sequentially, then the only change to the setting is that the fine-tuning step will be performed one by one for each class. We have performed a new experiment to see how this will change the results: https://tinyurl.com/y3tst9e2
>
> As the new results show, a sequential unlearning of multiple classes leads to slightly more deterioration in model’s utility, which is due to sequential I-projection of model to lower dimensional spaces as the fine-tuning process performs an approximate projection, and in the case of multiple classes, these projection errors accumulate.
>
> **References:** https://tinyurl.com/56x2tysc

---

> > ### Author Rebuttal · Reviewer_BB1z · 2026-04-03
> >
> > Thank the author for detailed response.

---

> > > ### Author Response · Authors · 2026-04-04
> > >
> > > Thank you again for your constructive feedback and for taking the time to review our responses.
> > >
> > > If you feel that your concerns have been sufficiently addressed, we would be grateful if you would consider updating your score to reflect this, as it can help provide a clearer signal to the AC regarding the contributions of our work.

---

### Decision · Program_Chairs · 2026-04-30

**Decision:**

Accept (regular)

**Comment:**

This paper introduces a new perspective on evaluating and executing Class Unlearning i.e. "forgetting" an entire category of data (like making a CIFAR-10 model forget the "Frog" class). Historically, unlearning methods were deemed successful if the model simply stopped predicting the forgotten class. However, the main observation is that if you train a model from scratch without the "Frog" class, the model doesn't just guess randomly when it sees a frog. Instead, it systematically reassigns that probability mass to semantically similar classes (like "Cat", "Bird", or "Deer"). Because most approximate unlearning methods just squash the "Frog" probability to zero and spread the remaining probability uniformly or randomly, they leave behind a detectable structural footprint. The authors exploit this footprint with their new Class Membership Inference Attack (CMIA) to prove these models haven't truly unlearned the class. The proposed solution here is  "Tilted REWeighting" which forces the unlearned model to mimic the geometric shift of a retrained model by explicitly "tilting" the reassigned probability mass toward the nearest neighbor classes.

Overall, all reviewers agreed that the Class Membership Inference Attack (CMIA) was a good contribution. They saw it as a clever and necessary tool that exposes a structural weakness in existing unlearning methods. On the other hand, reviewers had mixed opinions regarding the TREW algorithm. I agree with this criticism that the method is a heuristic and while it seems sensible it isn't in any sense optimal or rigorously obtained.

The reason I am recommending accept weakly and not strongly is the following. Class-wise unlearning is fundamentally a somewhat ill-defined problem and its not surprising that existing unlearning heuristics leak privacy under sufficiently strong MIA attacks (they never promised to protect against MIA anyway). So I am considering this paper to be a solid observation of an empirical issue and providing a clever empirical heuristic to protect against it. With that background, I would have expected a rigorous empirical evaluation to justify its usefulness (as also noted by multiple reviewers). However, reviewers and I found that lacking overall with small datasets like MNIST and CIFAR and relatively older MIAs. Thus I am hesitant to recommend a stronger acceptance.